# Cost-Effective Groundwater Potential Mapping by Integrating Multiple Remote Sensing Data and the Index–Overlay Method

Lamtupa Nainggolan [1,2,3], Chuen-Fa Ni [2,4,*], Yahya Darmawan [5], Wei-Cheng Lo [6], I-Hsian Lee [4], Chi-Ping Lin [4] and Nguyen Hoang Hiep [2]

1    Taiwan International Graduate Program (TIGP), Earth System Science Program, Academia Sinica, Taipei 11529, Taiwan; 107690608@cc.ncu.edu.tw
2    Graduate Institute of Applied Geology, National Central University, Taoyuan City 32001, Taiwan; 107684603@cc.ncu.edu.tw
3    Education and Culture Board of Medan City Government, North Sumatra 20233, Indonesia
4    Center for Environmental Studies, National Central University, Taoyuan City 32001, Taiwan; ihsienlee@ncu.edu.tw (I.-H.L.); chepinglin@g.ncu.edu.tw (C.-P.L.)
5    Indonesian College of Meteorology Climatology and Geophysics (STMKG), Banten 15221, Indonesia; yahya.darmawan@bmkg.go.id
6    Department of Hydraulic and Ocean Engineering, National Cheng Kung University, Tainan City 701, Taiwan; lowc@mail.ncku.edu.tw
*    Correspondence: nichuenfa@geo.ncu.edu.tw; Tel.: +886-3-4227151 (ext. 65874)

**Abstract:** The Choushui River groundwater basin (CRGB) in Yunlin County, Taiwan, is a significant groundwater source for the western part of the region. However, increasing groundwater demand and human activities have triggered a potential crisis due to overexploitation. Therefore, groundwater potential zone (GWPZ) maps are crucial for mapping groundwater resources and water resource management. This study employs the normalized index–overlay method and fuzzy extended analytical hierarchy process (FE-AHP) to map GWPZs cost-effectively. The methodology objectively incorporates weightings from various thematic layers by normalizing and correlating parameters with observed groundwater availability (GA). Site-specific observations, including aquifer thickness, depth to the groundwater level, and porosity, inform GA calculations. Seven comprehensive layers derived from remote sensing (RS) data are processed to obtain weightings and ratings for the groundwater potential index (GWPI) in the CRGB. Selected parameters are categorized into hydrological processes, human interventions, geological, and surface profiles. Hydrological processes include precipitation, modified normalized difference water index (MNDWI), and drainage density. Human interventions consist of the enhanced vegetation index (EVI) and normalized difference building index (NDBI). Surface profiles encompass the terrain ruggedness index (TRI) and slope, enhancing the study's multi-criteria approach. The observed GA validates the GWPZ accuracy, classifying zones into five categories. According to the GWPI of FE-AHP, about 59.56% of the CRGB area can be categorized as "moderate" to "very good" potential groundwater recharge zones. Pearson's correlation coefficient between GWPI and GA, based on FE-AHP, outperforms the conventional AHP. This RS-based approach efficiently evaluates GA in aquifers with limited wells, highlighting crucial zones in CRGB's proximal-fan and southeastern mid-fan for informed groundwater management strategies.

**Keywords:** index–overlay method; groundwater potential zone; fuzzy extended-AHP; groundwater availability

## 1. Introduction

In principle, the presence of groundwater depends on the interactions among several factors, such as hydrological, climatic, ecological, geological, and biological factors [1]. Identifying groundwater potential is essential for the local authorities to decide on a strategic plan for groundwater resource management in a vulnerable area. In most cases,

groundwater recharge is mainly influenced by soil permeability and porosity, which refer to the geological structure, geomorphological setting, lineaments, slope, land use, soil texture, and land use or land cover [1]. Site-specific groundwater drilling and stratigraphy investigations are two reliable methods for investigating aquifer properties, but these methods are relatively costly in terms of time and investigation resources [2]. In recent years, the integration of remote sensing and Geographical Information System (GIS) applications has been used widely in most hydrological studies. Groundwater potential pertains to the total volume of permanent storage within the initial layer of aquifers, often referred to as groundwater availability (GA) [3].

Groundwater potential is primarily determined by the porosity of rocks and the extent of open spaces within them capable of storing water [3]. Groundwater potential is principally affected by the porosity of rocks and favorable topography. Global groundwater storage constitutes the most voluminous fresh water available for human consumption. Conversely, the groundwater potential zone (GWPZ) encompasses a substantial and economically viable reservoir of groundwater resources, indicating a notable increase in groundwater availability [4]. Hence, investigating GWPZs is vital for estimating water resource reserves, zone budgeting, preservation of water quality, creating vulnerability maps, and effectively managing the environment. Most hydrogeological investigations and groundwater potential evaluations have traditionally been conducted using in situ measurements. However, these are not feasible in cases of limited funding or large area covers [3]. The latest improvements in GIS and RS technologies support advanced tools for groundwater monitoring and exploration [4]. Defining groundwater areas using GIS and RS becomes efficient and valuable in determining GWPZs [5].

However, feature classes used to define the GWPZs vary through cases, and the thematic map selection is subjective depending on the perspectives of different problems. The multi-criteria decision analysis (MCDA) method is widely used and suitable for complex and multi-criteria decision problems. Nevertheless, the potential of the MCDA technique in groundwater research is constrained by its limitation in accurately determining the appropriate weights for multi-thematic layers and their features. However, the MCDA technique is limited in potential groundwater research to choosing the correct weights for multi-thematic layers and the [5]. The AHP, proposed by Saaty (1980), can define the weights of multi-thematic layers with pairwise comparison [6]. However, there is a possibility of inconsistency at some stage of pairwise comparison in the AHP method. Then, Van Laarhoven and Pedrycz (1983) included fuzzy AHP by adopting the conventional AHP approach. They have involved the fuzzy set theory as an alternative for deriving the weight of each criterion [7]. In addition, fuzzy numbers are more realistic for defining the weights of multi-factors with subjective judgments.

At present, the AHP is often used to analyze multi-parameters because of its simplicity and effectiveness in dealing with complex decision-making problems [6]. However, in practice, a classical problem has to be addressed in the current situation about the uncertainty fuzziness and subjectivity in the conventional AHP, which makes the AHP method an inadequate tool for analyzing multi-layer data [8]. This limitation is overcome by integrating fuzzy logic into the AHP methodology, resulting in the FE-AHP. FE-AHP was proposed by Chang (1992) and Chang (1996) as an extension **of** the approach proposed by Saaty [6,9,10]. In principle, the FE-AHP method uses a triangular fuzzy number (TFN) during the fuzzification process [11]. By including the fuzzy matrix in the AHP method, the FE-AHP method can involve the human condition to integrate people's responses in decision making. As a result, the weighting of FE-AHP is closer to human reality than the conventional AHP method. Additionally, validations from previous studies indicated that the FE-AHP method held higher accuracy than the conventional AHP method based on expert validations [8].

Indeed, the AHP method has seen extensive application in recent decades and has proven successful in mapping groundwater potential zones [12–14]. The AHP is a conventional index–overlay method that can predict groundwater vulnerability in different climate

variations and human activities [15–17]. Applying the FE-AHP in mapping the GWPZ would improve the understanding of regional groundwater behavior in practical groundwater resource management. However, no previous studies have used the FE-AHP based on comprehensive inputs for mapping the GWPZ. The present study aims to integrate multiple RS data and employ the FE-AHP to delineate the GWPZ. Specifically, the developed FE-AHP and normalized index–overlay methods are applied to the Choushui River groundwater basin (CRGB) in Western Taiwan. The CRGB has been an area with intensive agriculture and aquaculture developments. There are sufficient groundwater monitoring stations to characterize spatial variations in shallow groundwater resources. Based on the site-specific monitoring data, the proposed FE-AHP and the associated GWPZ will be validated. The proposed approach could be useful for efficiently estimating groundwater availability in the first layer of an aquifer system, in which the groundwater monitoring data and aquifer properties are limited.

In the conventional AHP method, determining ranks for the AHP criteria requires expert judgments [18]. However, this study arranges the rank criteria based on the correlation matrix of the thematic maps versus the GA criteria. The correlation is a control variable for evaluating groundwater potential [3]. Specifically, the GA was derived based on observation data and will be used to validate the distribution of the final GWPI. In the present study, the AHP and FE-AHP were used to map groundwater potential zones to prove the concept of the proposed approach. For areas with limited groundwater monitoring systems, the GWPZ could provide input to policymakers and local authorities for effective and sustainable groundwater resource planning.

## 2. Materials and Methods

Mapping GWPZs is vital in addressing over pumping and promoting sustainable management for an aquifer system [19]. Specifically, they identify areas with high groundwater availability and recharge potential, helping the policymaker to locate suitable extraction sites [19]. This mapping provides valuable information on aquifers and vulnerability, enabling regulators to set pumping limits and develop long-term management strategies [20]. An accurate map of a GWPZ also helps identify areas at risk of over pumping, allowing targeted measures to prevent excessive extraction and mitigate negative impacts. For many areas where groundwater monitoring and hydrogeological data are unavailable, GWPZs calculated based on remote sensing data are efficient for groundwater resource planning and development. In summary, mapping GWPZs informs sustainable groundwater resource allocation, balancing water demand and mitigating possible overexploitation effects [20]. The processes of mapping GWPZs involve assessing the subsurface characteristics, hydrogeological conditions, and various factors that might influence the availability and quality of groundwater [21].

### 2.1. Study Area

The study area is in Yunlin County, Western part of Taiwan and known as the CRGB (see Figure 1a). The area of the CRGB is about 2500 km$^2$, bounded by Pakuashan tableland and Douliu Hill on the east side and the coastal line of the Taiwan Strait on the west. The regional surface and groundwater flow are from the east to the west along the gradually changed land surface from Pakuashan table land and Douliu Hill. The Choushui River, which passes the gap between Pakuashan tableland and Douliu Hill, develops the main river system in the study area. Drilling logs conducted in CRGB found the existence of various aquifers and aquitards from the Holocene to Pleistocene sands, gravels, and impermeable marine mud layers [22]. River and marine sedimentation processes have developed a complex system consisting of multiple aquifers and aquitards. The non-marine sequences were identified as aquifers. Because of the relatively high variation in the terrain slope, the aquifers consist of coarse sediment ranging from medium sand to gravel with high permeability. On the other hand, marine sequences comprising fine sediment ranging from clay to fine sand with low permeability are classified as aquitards in the CRGB [23].

In the context of the CRGB, the proximal-fan formation represents an unconfined aquifer. Aquitards are primarily situated in the distal-fan and mid-fan regions, distinct from the proximal-fan area. Based on the available logging data, the four aquifers are labeled "Aquifer 1" to "Aquifer 4" from top to bottom (Figure 1b).

The CRGB is an enormous alluvial plain in Central Western Taiwan. The alluvial fan is a crucial coastal aquifer that supports primary groundwater resources in the coastal region of Taiwan [24]. In the CRGB, groundwater dynamics are primarily influenced by direct precipitation and, to a certain extent, by local factors, as observed through river recharge [25]. With the non-uniform precipitation in time and space, the CRGB has regularly faced a shortage of surface water in dry seasons. Therefore, groundwater resources support the demand for various water uses in dry seasons, leading to high variations in groundwater levels in wet and dry seasons. Such high variations in groundwater levels are one of the key factors that trigger land subsidence in the CRGB. The land subsidence issue in the alluvial fan of the Choushui River in Western Taiwan has severe consequences for human-made structures, including railroads and buildings [26]. Previous investigations have proposed that extracting groundwater from areas with high groundwater potential could effectively balance the demand for groundwater resources and reduce land subsidence [27].

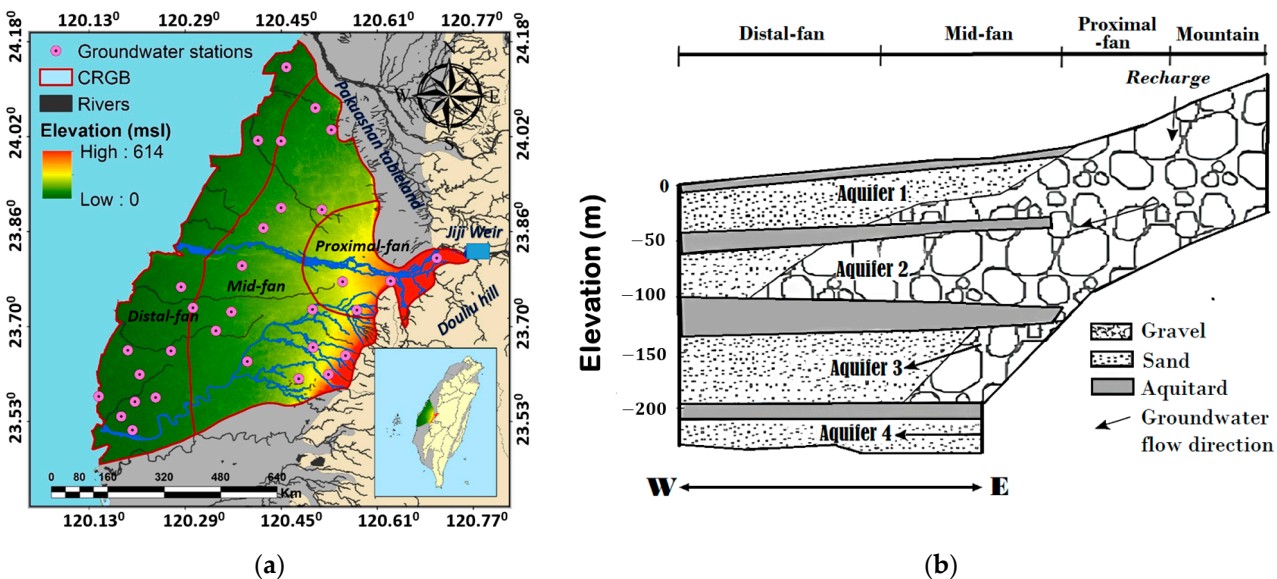

**(a)** **(b)**

**Figure 1.** (**a**) Choushui River groundwater basin (CRGB), overlaid by surface elevation data obtained from the SRTM, groundwater levels, and precipitation stations (CGS: WGS 1984) [28]; (**b**) the general hydrogeological profile for the Choushui River alluvial fan [27].

*2.2. Datasets*

This section explains the data sources and types utilized in the study. The logging data, which consist of collected lithology data, were obtained from the Central Geological Survey (CGS) in Taiwan. The monthly average groundwater levels from 2006 to 2015 were collected by the Taiwan Water Resource Agency (WRA) from 31 groundwater wells installed in the first aquifer layer. The yearly precipitation data from 2006 to 2015 were derived from the Climate Hazards Group InfraRed Precipitation with Station Data (CHIRPS) dataset. CHIRPS has a spatial resolution of 5.5 km, which was resampled to 30 m for this study. CHIRPS data can be downloaded for free from https://data.chc.ucsb.edu/products/CHIRPS-2.0/ (accessed on 15 October 2023). The enhanced vegetation index (EVI) was downloaded from the sensor of the MCD12Q1. It has been demonstrated in previous investigations to be a more reliable representation of the vegetation index in atmospheric disturbance compared to the normalized difference vegetation index (NDVI) [29]. The EVI can be downloaded from https://modis.ornl.gov/globalsubset/ (accessed on 15 October 2023) with a spatial resolution of 500 m. The land surface slope was obtained from the

Shuttle Radar Topographic Mission (SRTM) dataset with a spatial resolution of 30 m, accessible at https://earthexplorer.usgs.gov/ (accessed on 15 October 2023). The modified normalized difference water index (MNDWI) was derived from cloud-free Landsat 8 OLI (Operational Land Imager) Level 2 satellite images taken in 2015 and obtained from the United States Geological Survey (USGS). These images can also be accessed at https://earthexplorer.usgs.gov/ (accessed on 15 October 2023). Table 1 presents the Landsat 8 OLI/TIRS data bands, wavelengths, and resolutions used in the study. Multiple software tools, including Matlab R2016b, ArcGIS 10.8, and the R 4.2.3 package, were utilized for data processing.

**Table 1.** Landsat 8 OLI/TIRS data bands, wavelength, and resolution for the study.

| Band | Landsat 8 Operational Land Imagers (OLIs) and Thermal Infrared Sensor (TIRS) | | |
| --- | --- | --- | --- |
| | **Band Name** | **Wavelength (Micrometers)** | **Resolution (Meters)** |
| Band 1 | Ultra-Blue | 0.435–0.451 | 30 |
| Band 2 | Blue | 0.452–0.512 | 30 |
| Band 3 | Green | 0.533–0.590 | 30 |
| Band 4 | Red | 0.636–0.673 | 30 |
| Band 5 | NIR | 0.851–0.879 | 30 |
| Band 6 | SWIR 1 | 1.566–1.651 | 30 |
| Band 7 | SWIR 2 | 2.107–2.294 | 30 |
| Band 8 | Panchromatic | 0.503–0.676 | 15 |
| Band 9 | Cirrus | 1.363–1.384 | 30 |
| Band 10 | TIRS 1 | 10.60–11.19 | 100 ∗ (30) |
| Band 11 | TIRS 2 | 11.50–12.51 | 100 ∗ (30) |

*2.3. Methods*

Figure 2 presents the general flowchart in this study to map the GWPZ. The GWPZ was determined by calculating the GWPI for the entire groundwater basin [25]. The GWPI represents the complex interplay of socioeconomic factors, hydrometeorology, topography, and land resources [26]. A GWPZ map employs a weighted index overlay concept, where weight values are assigned to each thematic layer [27]. As shown in the flowchart of Figure 2, the study determined the GA based on the collected site-specific data, including the aquifer thickness of the first layer, groundwater depths, and the porosity of the shallow aquifer. The results of the GA were for the validation of the GWPZ. We then conducted the Band Collection statistics for normalized thematic maps. The index–overlay method requires weightings and ratings for each specific thematic map. We utilized the FE-AHP to assess the thematic layers and define their weightings. The ratings and weightings of the selected parameters were used to determine the GWPI for each cell in the map. Subsequently, the GWPZ map was generated based on the GWPI values.

The study employed the direct calculations of groundwater volume in the first layer (i.e., the GA) and compared the GA with the obtained GWPZ to assess the accuracy of the GWPZ map. A Pearson correlation matrix between the GA and GWPZ was built to evaluate the linear relationship between the GA and GWPZ. This step aimed to examine the correlation between the GWPZ and the actual groundwater availability. On-site field verifications were also conducted by checking the selected high and low GWPZ areas in the CRGB. A Google Street map with high-resolution satellite images was able to provide specific references for the selected sites [29].

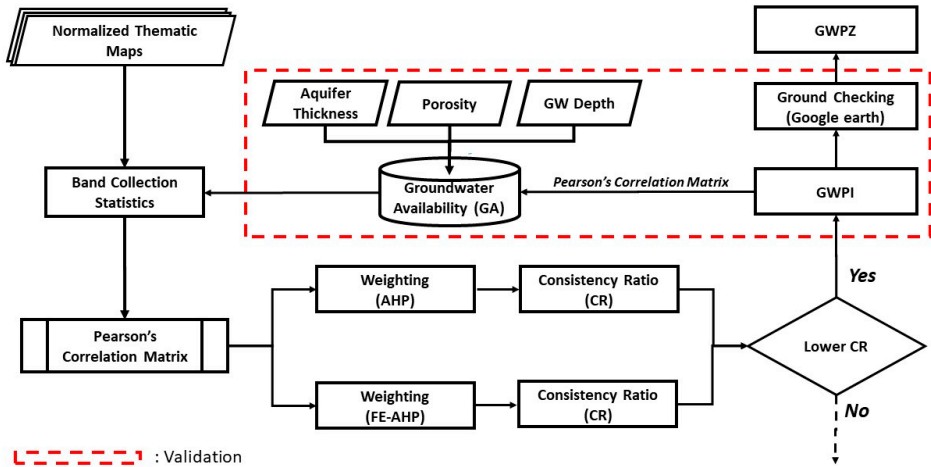

**Figure 2.** The flowchart for mapping and validating the groundwater potential zone (GWPZ).

2.3.1. Groundwater Availability (GA)

In this study, groundwater availability accounted for the storage potential in the pore space of the shallow aquifer. Therefore, calculating the groundwater availability considered site-specific observations such as the aquifer thickness, depth to the groundwater level, and porosity of the aquifer materials. To quantify the groundwater availability, we calculated the groundwater volume within the pore space of the saturated portion in the first aquifer layer. It was calculated by subtracting the groundwater depth from the aquifer thickness of the shallow aquifer and then multiplying the result by the porosity. Additionally, the flatness of the topography showed a noteworthy impact on lateral inflows, also known as the percolation rate. We considered the monthly averaged groundwater depth to identify seasonal groundwater fluctuations in the study area. February was characterized as a dry month and was selected to represent the minimum amount of precipitation received across the study area. Due to limited porosity data, a porosity map was derived based on the identified soil texture and integrated with the USDA survey database [30] (see Table 2). The common porosity values in Table 2 were used in the study. For a specific location, higher groundwater availability could indicate a higher GWPI (groundwater potential index) and vice versa. In the study area, the calculation of the groundwater availability used the February groundwater levels obtained from 2006 to 2015.

**Table 2.** Porosity based on soil texture from Clapp and Hornberger (1978).

| Soil Texture | Porosity [-] |
| --- | --- |
| Sand | 0.395 |
| Loamy sand | 0.410 |
| Sandy loam | 0.435 |
| Silt | 0.485 |
| Loam | 0.451 |
| Sandy clay loam | 0.420 |
| Silty clay loam | 0.477 |
| Clay loam | 0.476 |
| Sandy clay | 0.426 |
| Silty clay | 0.492 |
| Clay | 0.482 |

2.3.2. Analytic Hierarchy Process (AHP)

The AHP, proposed by Saaty [6], has found extensive application in multi-criteria evaluation for decision-making scenarios involving conflicting and qualitative criteria. The AHP involves a structured approach of pairwise comparisons, utilizing a standardized nine-level scale to determine the relative importance of criteria. The AHP involves a structured

set of pairwise comparisons, utilizing a standardized nine-level scale to evaluate the relative importance of criteria. Each criterion is assigned ratings or weights based on these comparisons, using values ranging from 1 to 9, reflecting the degree of importance (from extremely less important to extremely more influential). The impact of factors or criteria on the decision-making process is quantified by combining this scale with the expertise and knowledge of specialists or users [31]. The AHP method allows for a comprehensive consideration of both subjective and objective evaluation measures while also providing a means to test the stability of evaluation methods and proposed options through specialists or decision-makers, thus reducing errors in decision making [31]. This study employed the results of the classical AHP for comparison purposes. Specifically, the maps of the GWPZs obtained from the conventional AHP and the FE-AHP were quantitatively evaluated and assessed.

Most studies utilizing multi-criteria evaluation, such as the AHP, rely on expert judgment to rank the criteria. However, this study introduced a novel approach using a correlation matrix between selected thematic maps derived from seven RS data and an independent variable not included in the model. The study considered GA as the independent variable strongly correlated with the GWPI [3]. In AHP methods, the initial step involves tabulating the data based on Pearson's correlation matrix built between the normalized thematic maps and groundwater availability. The correlation matrix among the thematic layers was generated using Band Collection Statistics in ArcGIS 10.8. This method allows for ranking each criterion of the GWPZ, providing a more objective approach than relying solely on expert judgment or the existing literature. Here, the z-score normalization or zero mean normalization method was employed to normalize the data. This normalization method involves subtracting the mean ($\mu$) of each feature and dividing it by the standard deviation ($\sigma$) [27].

$$v'_i = \frac{v_i - \mu}{\sigma} \tag{1}$$

where $v_i$ is the value of the criteria for every grid of $i$, $\mu$ is the expectation of the variable, $\sigma$ is the standard deviation of the variable, and the notation $v'_i$ represents the z-score of the corresponding outlier. Reference data are presented in the form of seven data on the GWPI, which were ranked as reference partners, along with seven thematic maps. The seven thematic maps were then determined by considering the analysis of the needs and availability of RS data. However, the data types could be changed based on different situations and site-specific conditions. In this study, the selected parameters were based on the hydrology processes, human interventions, geological profile and surface profile [11]. For hydrological processes, they are characterized by specific parameters such as P (precipitation in mm), MNDWI (modified normalized difference water index), and DD (drainage density in km/km$^2$). Human interventions are identified by parameters, namely, the EVI (enhanced vegetation index) and NDBI (normalized difference building index). Geological profile and surface profile are explained by TRI (terrain ruggedness index) and SL (slope in degrees), respectively.

Note that the AHP involves breaking down the issue into a hierarchy of elements, specifically the criteria and the sub-criteria. In this study, the criteria consisted of the seven parameters, while the sub-criteria were the different ranges within each parameter.

Precipitation (P). Precipitation is the primary source for groundwater recharge in aquifer systems [11]. Hence, increasing precipitation represents the increasing groundwater potential over a specific area. Precipitation is closely related to the amount of surface water that infiltrates and percolates into the aquifer as the input for the groundwater recharge [32]. As a result, a variation in the spatial intensity of precipitation is referred to as the variation in groundwater recharge rate across CRGB. The yearly precipitation data from 2006 to 2015 were derived from the Climate Hazards Group InfraRed Precipitation With Station Data (CHIRPS) dataset.

Drainage density (DD). The drainage density is the ratio of the stream segments in a specific area. Drainage is one of the crucial parameters in hydrogeological processes, which

controls the interactions near land surfaces. In an aquifer system, the drainage density could significantly influence groundwater recharge. Lower groundwater infiltration happens in the media with higher drainage density. A higher drainage density associated with the lower soil permeability on land surfaces could lead to lower water infiltration and higher surface runoff. In the study, we employed the available line density algorithm in ArcGIS to calculate the drainage density in CRGB. Drainage density with the unit of km/km$^2$ represents the closeness among the stream channels.

Enhanced vegetation index (EVI). Areas with dense vegetation are hydrologically more stable due to their typically better soil infiltration properties, attributed to higher organic matter content. Different vegetation types respond differently to groundwater presence in aquifers [33]. NDVI is a conventional vegetation index, but recent studies suggest that the EVI, with a more robust profile in areas with atmospheric disturbance, can be derived from Landsat 8–9 data [34]. Equation (2) outlines the general formula for determining the EVI, covering the canopy background with the L value. Coefficients for atmospheric corrections and the blue band (B) are represented by C values, helping reduce noise induced by background, atmospheric conditions, and saturation in cloud-covered areas.

$$EVI = G * \frac{(NIR - Red)}{(NIR + C1 * Red - C2 * Blue + L)} \qquad (2)$$

In Landsat 8–9, the EVI for the study area can be calculated by $2.5 * ((\text{Band } 5\text{–Band } 4)/(\text{Band } 5 + 6 * \text{Band } 4\text{–}7.5 * \text{Band } 2 + 1))$. The EVI has a range between $-1$ and 1. When an area has an EVI closer to 1, then the vegetation over that area is very dense and has a higher value for groundwater potential, and vice versa.

Modified normalized difference water index (MNDWI). The MNDWI was applied to detect the water bodies on the land surface [35]. The objective of the MNDWI is to reduce the effect of features in the built-up areas that are often detected together with open water like other indices. The algorithm of the MNDWI can be seen in Equation (3):

$$MNDWI = \frac{(Green - SWIR1)}{(Green + SWIR1)} \qquad (3)$$

Moreover, in the case of the modified normalized difference water index (MNDWI), the pixel values extracted from the Green (3rd band) and short-wave infrared SWIR1 (6th band) play a pivotal role. The synergistic utilization of these specific bands from the Landsat 8–9 satellite imagery facilitates a comprehensive analysis of water bodies, aiding in accurately delineating and characterizing aquatic features within the study area.

Terrain ruggedness index (TRI). Topography is essential in controlling the spatial variability of hydrological processes such as surface and groundwater flow and soil moisture. Topographic indices have been applied to represent the spatial pattern of soil moisture [36]. The topographic ruggedness index to quantify the elevation difference between adjacent cells using DEM obtained from SRTM [36]. Terrain roughness, such as micro-relief, terrain rugosity, ruggedness, surface roughness, and micro topography, can be defined as the variation in elevation. A higher TRI value of a pixel translates to a more considerable difference in altitude compared to the adjacent areas around that pixel.

Slope (SL). Slope factors intensely affect the lateral and vertical flow of groundwater [33]. Previous investigations have recognized that the land surface slope considerably controls the infiltration rate of surface water, which is mainly related to the groundwater recharge of an aquifer system (e.g., Ref. [37]). In principle, the percolation rate of surface water has a negative linear interaction with the land surface slope because of the retention time of the surface runoff. The surface runoff is relatively lower in a gentle slope area than in a sloping area. Therefore, a gentle slope will lead to a higher rate of percolation.

Normalized difference building index (NDBI). The conventional image classification technique is usually used to classify satellite images based on supervised and unsupervised classification methods. However, these methods are ineffective, including steps with

complex procedures. Specifically, the operations require composite bands and judgment parameters for the final results. The NDBI technique is more effective than conventional classification methods. The reflectance for built-up areas and bare lands is relatively higher for SWIR than for NIR. For a green surface, the reflection of NIR is higher than that of the SWIR spectrum. In contrast, water bodies cannot be detected by the infrared spectrum. The NDBI gives the following formula (Equation (4)):

$$NDBI = \frac{(SWIR - NIR)}{(SWIR + NIR)} \tag{4}$$

For Landsat 8 data, the NDBI can be calculated using the formula (Band 6–Band 5)/ (Band 6 + Band 5). Also, the NDBI has a value range between −1 and +1. A negative value of NDBI represents water bodies, whereas a positive value represents built-up areas. Identifying water bodies is essential to indicate the possible recharge zones for an aquifer system. Note that the NDBI value for vegetation is generally lower than the NDBI value for water (see Tables A13 and A14 for details) [38].

The AHP requires constructing judgment matrices ($B_w$) of size ($n \times n$) through pairwise comparisons among the n criteria. The diagonal elements are all set to one in these matrices since they represent the same criterion. Subsequently, the relative weights for each matrix are determined by identifying the right eigenvector ($w$) corresponding to the largest eigenvalue ($\lambda_{max}$) (Equation (5)).

$$B_w = \lambda_{max}w \tag{5}$$

As the value of $\lambda_{max}$ approaches the number of elements ($n$) in the pairwise comparison matrix, the judgments in the matrix become increasingly consistent. Hence, the difference, $\lambda_{max} - n$, can be used as an indicator of inconsistency. To assess the coherence of the judgments, Saaty [30] introduced a consistency index ($CI$) to measure the agreement among the $B$ matrices, where $b_{ij} \times b_{jk} = b_{ik}$. The formula for calculating the consistency index can be expressed as (Equation (6)):

$$CI = \frac{\lambda_{max} - n}{n - 1} \tag{6}$$

Random pairwise comparisons on matrices of varying sizes determine the consistency index. The random index ($RI$) can be calculated by averaging the consistency indexes for each matrix size. The consistency ratio ($CR$) is subsequently defined as the ratio between the consistency index and the random index ($RI$) (Equation (7)).

$$CR = \frac{CI}{RI} \tag{7}$$

A $CR$ value exceeding 0.1 indicates a significant inconsistency in the judgments made during the creation of the pairwise comparison matrix. Therefore, it is necessary to maintain a $CR$ at ≤0.1 to ensure the stability of the array. The $CR$ measures how consistent the pairwise comparisons are in the AHP analysis. In addition, the $CR$ is used to evaluate the reliability of the judgments made during the comparison process. Note that the significance of the difference between these two consistency ratios depends on the context and the specific threshold used in the model. In general, the lower the consistency ratio, the more reliable the judgments made in the analysis [39].

### 2.3.3. Fuzzy Extended AHP (FE-AHP)

Zadeh [11] introduced the fuzzy set theory in 1965. The study showed that using the membership functions by real numbers [0, 1] is acceptable. The process is called generalization for a classic set theory. In principle, the primary characteristic of fuzziness is individuals grouping into some classes. At that point, it allows unclear boundaries or bias for the threshold of each class [40]. Then, the ambiguous comparison of the judgment can be characterized by fuzzy numbers. The TFN is a unique class of fuzzy numbers in

which three real numbers define the membership: (*l*, *m*, *u*). A TFN is symbolized by (*l*, *m*, *u*), where *l*, *m*, and *u* refer to the smallest, most promising, and largest possible values, respectively (Equation (8)). In some cases, it could occur when the data are difficult to specify precisely because of measurement or instrument error. However, an accurate height measurement is rarely obtained in practice, with it usually slightly more or slightly less than the real value. Thus, the measurement numbers can be written more accurately as the TFN (Table 3). Figure 3a shows the conceptual structure of the TFN of the conventional AHP method [41].

$$\mu_A = \begin{cases} (x-l)/(m-l), & l \le x \le m \\ (u-x)/(u-m), & m \le x \le u \\ 0 & otherwise \end{cases} \tag{8}$$

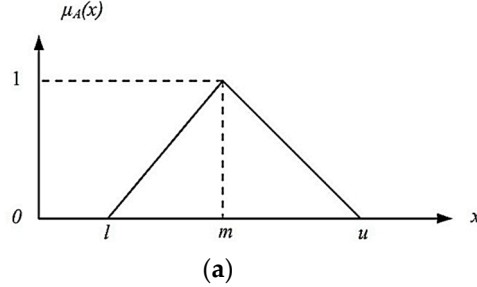

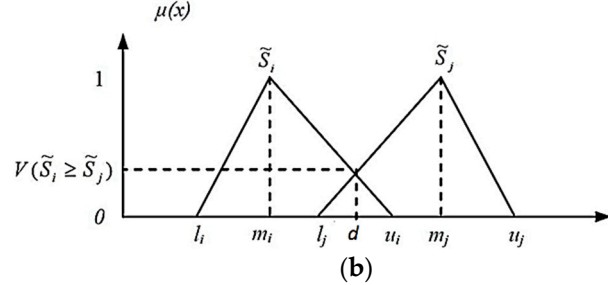

(**a**)　　　　　　　　　　　　　　　　(**b**)

**Figure 3.** Triangular membership function (TFN) and the intersection between AHP and FE-AHP. The structure of TFN between AHP and FE-AHP can be listed as follows: (**a**) the structure of the TFN for conventional AHP; (**b**) the degree of possibility of $V\left(\tilde{S}_i \ge \tilde{S}_j\right)$ for FE-AHP, where the intersection point "*d*" is between two fuzzy numbers $\tilde{S}_i$ and $\tilde{S}_j$ (modified from [42]).

To create a pairwise comparison of alternatives for each criterion, similar to the concept of the conventional AHP method, a matrix of triangular fuzzy comparison can be defined as follows (Equation (9)):

$$\tilde{A} = \left(\tilde{a}_{ij}\right)_{n \times n} = \begin{bmatrix} (1,1,1) & (l_{12}m_{12}u_{12}) & \cdots & (l_{1n}m_{1n}u_{1n}) \\ (l_{21}m_{21}u_{21}) & (1,1,1) & \cdots & (l_{2n}m_{2n}u_{2n}) \\ \vdots & \vdots & \vdots & \vdots \\ (l_{n1}m_{n1}u_{n1}) & (l_{n2}m_{n2}u_{n2}) & \cdots & (1,1,1) \end{bmatrix} \tag{9}$$

where $\tilde{a}_{ij} = \left(l_{ij}m_{ij}u_{ij}\right) = \tilde{a}_{ij}^{-1} = \left(1/u_{ji}, 1/m_{ji}, 1/l_{ji}\right)$ for $i, j = 1, \ldots n$, and $i \ne j$.

The total weights and preferences of alternatives could be developed from other methods. In general, two alternative methods will be modeled in continuation. Figure 3 summarizes the conceptual structure of the conventional AHP and FE-AHP. The conventional AHP uses a TFN to represent linguistic variables and capture imprecise judgments. It consists of three parameters, including *l* (lower bound), *m* (modal value), and *u* (upper bound) (see Figure 3a). TFNs are commonly used in the AHP to express linguistic terms, such as "low", "medium", and "high", or to represent uncertainty in pairwise comparisons.

The FE-AHP method was proposed by Chang (1996) [10,11]. The extended analysis of Chang (1996) can be concisely described in several steps. First, compute the normalized value for row sums by fuzzy arithmetic operations (Equation (10)):

$$\tilde{S}_i = \sum_{j=1}^{n} \tilde{a}_{ij} \otimes \left[\left|\sum_{k=1}^{n}\sum_{j=1}^{n} \tilde{a}_{kj}\right|\right]^{-1} \tag{10}$$

where the notation $\otimes$ represents the extended multiplication of two fuzzy numbers. Second, compute the degree of possibility using the following (Equation (11)):

$$V\left(\tilde{S}_i \geq \tilde{S}_j\right) = \sup_{y \geq x}\left[\min\left(\tilde{S}_j(x), \tilde{S}_i(y)\right)\right] \tag{11}$$

which can be equally written by (Equation (12)):

$$V\left(\tilde{S}_i \geq \tilde{S}_j\right) = \text{hgt}\left(\tilde{S}_i \cap \tilde{S}_j\right) = \tilde{S}_j(d) = \begin{cases} 1 & m_i \geq m_j \\ \frac{u_i - l_j}{(u_i - m_i) + (m_j - l_j)} & l_j \leq u_i \\ 0 & \text{otherwise} \end{cases} \quad i,j = 1,\ldots,n; j \neq i \tag{12}$$

where $\tilde{S}_i = (l_i m_i u_i)$ and $\tilde{S}_j = (l_j m_j u_j)$, and d is the ordinate of the highest intersection point between $\mu_{\tilde{S}_i}$, $\mu_{\tilde{S}_j}$ (see Figure 3b).

The FE-AHP method is an improved version of the AHP that introduces additional features to handle more complex decision problems involving uncertainties. The FE-AHP method incorporates fuzzy numbers to represent imprecise information, but instead of using TFNs, it uses fuzzy number intervals (Figure 3b). The representation of fuzzy numbers in the FE-AHP is more flexible and allows for a more extensive range of uncertainty [43].

In the FE-AHP, a fuzzy number is represented as $\tilde{S}_i = (l_i m_i u_i)$, where $(l_i m_i u_i)$ are lower, modal, and upper values, respectively. Similarly, another fuzzy number $\tilde{S}_j = (l_j m_j u_j)$ can be defined (Figure 3b). The FE-AHP allows an intersection point "d" between two fuzzy numbers $\tilde{S}_i$ and $\tilde{S}_j$. These fuzzy number intervals allow decision-makers to express their preferences and judgments more detailed and nuancedly [43]. The FE-AHP also introduces the concept of fuzzy pairwise comparison matrices, which capture the imprecise judgments between criteria or alternatives. These matrices use fuzzy numbers or fuzzy number intervals to accommodate uncertainty and imprecision during the decision-making process.

Third, calculate the degree of possibility of $\tilde{S}_i$ to be larger than all the other (n − 1) convex fuzzy numbers $\tilde{S}_j$ (Equation (13)):

$$V\left(\tilde{S}_i \geq \tilde{S}_j \middle| j = 1,\ldots,n; j \neq i\right) = \min_{je(1,\ldots,n)j \neq i} V\left(\tilde{S}_i \geq \tilde{S}_j\right), \ i = 1,\ldots,n \tag{13}$$

Fourth, define the vector of priority $W = (w_1,\ldots,w_2)^{\Gamma}$ for the fuzzy comparison matrix $\tilde{A}$ as (Equation (14)):

$$w_i = \frac{V\left(\tilde{S}_i \geq \tilde{S}_j \middle| j = 1,\ldots,n; j \neq i\right)}{\sum_{k-1}^{n} V\left(\tilde{S}_i \geq \tilde{S}_j \middle| j = 1,\ldots,n; j \neq k\right)} i = 1,\ldots,n \tag{14}$$

In summary, both TFNs for the AHP and FE-AHP methods involve fuzzy numbers. The TFN relies on a triangular representation (*l*, *m*, *u*) to express imprecise judgments, whereas FE-AHP employs fuzzy number intervals $(l_i m_i u_i)$, to provide a more versatile representation for capturing uncertainties and complex decision problems. In the FE-AHP method, the degree of possibility is suggested for the ordering and the weights. In this step, a pairwise comparison is made for every fuzzy weight by other fuzzy weights. The conforming degree of possibility of being higher than other fuzzy weights is defined. The minimum of the possibility is used as the overall score for each criterion.

**Table 3.** The triangular fuzzy number of the FE-AHP method [44].

| AHP Scale | Linguistic Variable | FE-AHP Scale | |
|:---:|:---:|:---:|:---:|
| | | TFN Number | Reciprocal |
| 1 | Equally important | (1, 1, 1) | (1, 1, 1) |
| 2 | Intermediate of 1 to 3 | (1/2, 1, 3/2) | (2/3, 1, 2) |
| 3 | Slightly important | (1, 3/2, 2) | (1/2, 2/3, 1) |
| 4 | Intermediate of 3 to 5 | (3/2, 2, 5/2) | (2/5, 1/2, 2/3) |
| 5 | Important | (2, 5/2, 3) | (1/3, 2/5, 1/2) |
| 6 | Intermediate of 5 to 7 | (5/2, 3, 7/2) | (2/7, 1/3, 2/5) |
| 7 | Strongly important | (3, 7/2, 4) | (1/4, 2/7, 1/3) |
| 8 | Intermediate of 7 to 9 | (7/2, 4, 9/2) | (2/9, 1/4), 2/7) |
| 9 | Extremely important | (4, 9/2, 9/2) | (2/9, 2/9, 1/4) |

### 2.3.4. Groundwater Potential Index (GWPI)

The groundwater potential index (GWPI) is a dimensionless and quantification index used to calculate potential groundwater scores in various areas by integrating thematic layers and the corresponding weightings and ratings [28]. The GWPI can be determined by integrating these thematic layers into a Geographic Information System (GIS) platform. The ratings and weights of each class are defined using the fuzzy evaluation analytic hierarchy process (FE-AHP) method. The GWPI can be calculated using the following equation (Equation (15)):

$$\text{GWPI} = [(\text{Pr} * \text{Pw}) + (\text{DDr} * \text{DDw}) + (\text{EVIr} * \text{EVIw}) + (\text{MNDWIr} * \text{MNDWIw}) + (\text{TRI} * \text{TRIw}) + (\text{SLr} * \text{SLw}) + (\text{NDBIr} * \text{NDBIw})] \tag{15}$$

The equation for calculating the GWPI involves various parameters represented by subscripts "r" (rating) and "w" (weight). These parameters include P for precipitation (mm), DD for drainage density (km/km$^2$), EVI for the enhanced vegetation index, MNDWI for the modified normalized difference water index, TRI for the terrain ruggedness index, SL for slope (degree), and NDBI for the normalized difference building index. The GWPI was derived from seven thematic maps of RS data to generate the map for GWPZ. Each thematic datum was converted into raster datasets using ArcGIS software. The GWPI was calculated using the index–overlay method and used to create the final GWPZ map. The weighting factors for all criteria were defined based on the location properties shown in Table 3. The GWPZ map was divided into five classes, including very high, high, moderate, poor, and very poor, using the natural break (Jenks) classification method. This method optimizes the classification of values into different classes and provides a relative probability assessment of groundwater potential resources [45]. Natural break (Jenks) is one of the clustering data methods. This method uses an optimization process to find the best value classification into different classes [45].

### 2.3.5. Validation of the GWPZ

First, the GWPZ map was validated using Pearson's correlation matrix between the GWPI and the direct calculation of the GA for the first aquifer layer in the CRGB. Note that the GA was not included in the analysis of the GWPZ. The GA served as an independent parameter that robustly confirmed the accuracy of the GWPZ. Initially, the GA was solely utilized to determine the ranking and was not included in the GWPZ map. For raster datasets, the correlation matrix provides cell values from one raster layer to another. This correlation between layers enables the measurement of the degree of dependency between them. Correlation values range from +1 to −1. A positive correlation indicates a direct relationship between the layers, whereas a negative correlation signifies an inverse relationship where the variables change in opposite directions. The study also conducted validation based on ground checking. Google Earth provides satellite imagery with varying spatial and temporal resolution [46]. The spatial resolution of Google Earth imagery varies

depending on location and data availability. In densely populated areas or popular tourist destinations, the spatial resolution tends to be higher, enabling more detailed views of buildings, streets, and landmarks [46]. Conversely, remote or less frequented areas may have a lower spatial resolution, resulting in less detailed information [46].

Second, delineating groundwater recharge potential zones involves a weighted overlay of different themes in the geospatial environment. This method serves as an indirect measurement to identify potential zones but requires validation through direct observations for accurate planning. The validation process is a crucial element in scientific research. A random sampling technique is employed in the geospatial environment to validate the groundwater recharge potential zones with actual recharge. Using Google Earth, multiple locations were selected randomly to represent each class of the GWPZ, allowing for a comprehensive assessment of the GWPZ's accuracy. As a model consistency validation, locations were selected based on our knowledge of their real or physical profile. For instance, the downstream area in Mailiao Haipu New Land, western Yunlin County, features diverse land uses, including coastal uplift rejuvenation, siltation sites, saline ponds, aqua fisheries, and sandbanks affected by monsoonal activity. Hypothesizing, Mailiao Haipu New Land is expected to have a poor geopotential zone due to salty water and seawater intrusion. Conversely, the upstream area includes agriculture and water resources, indicating a good groundwater potential zone.

### 3. Results and Discussion

*3.1. Estimation of the Groundwater Availability (GA)*

The GA was calculated to determine the amount of water available in each cell of the first aquifer layer. Figure 4 shows the result of the calculated GA. High GA was predominantly found in the proximal-fan area in the study area. This observation agrees with previous research indicating that the proximal-fan has a higher potential for groundwater recharge [47,48]. A higher groundwater recharge potential rate corresponds to a higher likelihood of groundwater potential zones. Based on Pearson's correlation matrix, the parameter with the highest correlation to GA was precipitation, followed by drainage density, EVI, MNDWI, TRI, slope, and NDBI (Table 4). Notably, a negative relationship was observed between groundwater availability, MNDWI, and NDBI. Groundwater recharge primarily occurs through the infiltration process of precipitation. In a regional groundwater system, the proximal-fan, located in the upstream area with the highest elevation, serves as the main recharge zone for the groundwater resource. The higher correlation between precipitation and GA compared to other parameters can be attributed to the fact that the Choushui River carries most of the groundwater resources originating from the precipitation. The correlation r between GA and the precipitation was 0.68 for the study. Following the precipitation, the second important parameter with a relatively strong correlation to GA was the drainage density (DD), with a Pearson's correlation of $-0.66$. The EVI showed a slightly low correlation with GA compared to the correlation between precipitation and GA (r = 0.64). In this model, EVI was used as a representation of vegetation cover. Previous research has established a significant linear relationship between groundwater levels and vegetation cover [49]. The presence of vegetation cover will support the percolation and infiltration of water.

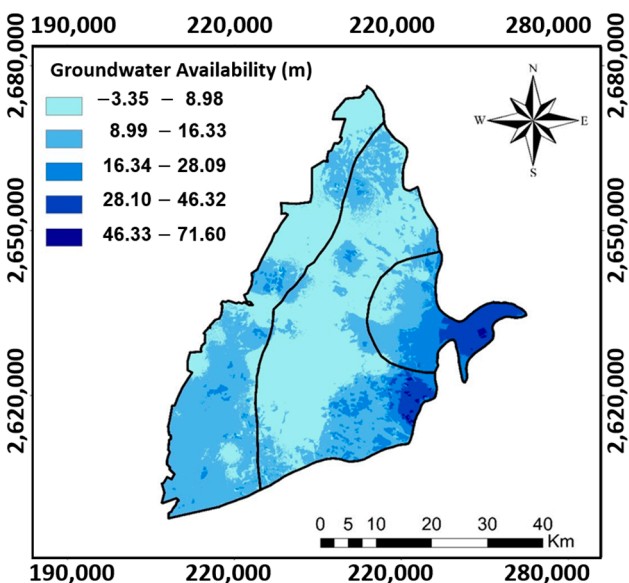

**Figure 4.** The calculated groundwater availability (GA) based on the site-specific observations (PCS: UTM zone 51N).

**Table 4.** Correlation matrix between GA and thematic layers.

| Layers | GA | P | DD | EVI | MNDWI | TRI | SL | NDBI |
|--------|------|-------|-------|-------|-------|-------|-------|-------|
| GA | 1.00 | 0.68 | −0.66 | 0.64 | −0.62 | −0.52 | −0.50 | 0.48 |
| P | 0.68 | 1.00 | −0.61 | 0.77 | 0.60 | 0.65 | 0.65 | −0.59 |
| DD | −0.66 | −0.61 | 1.00 | −0.39 | −0.39 | −0.39 | −0.39 | −0.39 |
| EVI | 0.64 | 0.77 | −0.39 | 1.00 | −0.76 | −0.76 | −0.76 | −0.76 |
| MNDWI | −0.62 | 0.60 | 0.50 | −0.76 | 1.00 | 0.38 | 0.38 | 0.38 |
| TRI | −0.52 | 0.65 | 0.50 | −0.43 | 0.38 | 1.00 | 0.61 | 0.61 |
| SL | −0.50 | 0.65 | 0.49 | −0.45 | 0.20 | 0.61 | 1.00 | −0.36 |
| NDBI | 0.48 | −0.59 | 0.33 | −0.73 | 0.23 | −0.42 | −0.36 | 1.00 |

### 3.2. Thematic Layers of the GWPZ

We considered seven reference variables to calculate the GWPI for the CRGB. These variables include precipitation (P), drainage density (DD), enhanced vegetation index (EVI), modified normalized difference water index (MNDWI), terrain ruggedness index (TRI), slope (SL), and normalized difference building index (NDBI). These variables are determined based on the availability of the RS data, which may vary depending on site-specific conditions. By including multiple parameters, the weighting of the factors is less likely to be dominated by a specific criterion, allowing for a more comprehensive representation of the system complexity. Table 5 summarizes the weightings and ratings obtained from conventional AHP and FE-AHP analyses. The detailed calculations of the weightings for the AHP and FE-AHP are listed in Appendix A.

The AHP and FE-AHP utilize the GA to rank and arrange the thematic parameters in the study. After the parameter arrangement was completed, the data were used to calculate the AHP and FE-AHP. However, the notable difference between the AHP and FE-AHP is that despite using a similar scale, the FE-AHP generated more reasonable weights between the criteria and smoother ratings. The differences in weightings and ratings are not as contrasting in the FE-AHP compared to the AHP, allowing for a smoother transition in the FE-AHP [39]. For example, in the AHP, the weight difference between all criteria and the rating within a criterion is more contrasting and rough. One instance is the precipitation rating for sub-parameters in the interval 2088–2547, which gives the value of 0.4537, followed by a rating of 0.2667 for the interval 1830–2087. However, the FE-AHP assigns ratings of 0.3870 and 0.3042 for similar sub-parameters, resulting in a more

gradual and smoother change. In summary, while both the AHP and FE-AHP employ GA for parameter ranking, the FE-AHP demonstrates a more reasonable weight distribution between criteria and smoother transitions in ratings compared to the conventional AHP. This characteristic of the FE-AHP contributes to a more refined and consistent decision-making process (see Table 5).

**Table 5.** Classification of thematic layers of the GWPZs in the CRGB based on the AHP and FE-AHP methods.

| Parameters | Sub-Parameters | AHP | | FE-AHP | |
|---|---|---|---|---|---|
| | | Weight | Rating | Weight | Rating |
| P | 1117–1442 | | 0.0488 | | 0.0171 |
| | 1443–1616 | | 0.0817 | | 0.0909 |
| | 1617–1829 | 0.3354 | 0.1491 | 0.2661 | 0.2008 |
| | 1830–2087 | | 0.2667 | | 0.3042 |
| | 2088–2547 | | 0.4537 | | 0.3870 |
| DD | 0–1.71 | | 0.4456 | | 0.3870 |
| | 1.72–3.06 | | 0.2690 | | 0.3042 |
| | 3.07–4.23 | 0.2320 | 0.1512 | 0.2277 | 0.2008 |
| | 4.24–5.53 | | 0.0827 | | 0.0909 |
| | 5.54–9.56 | | 0.0514 | | 0.0171 |
| EVI | −0.29–(−0.01) | | 0.0588 | | 0.093 |
| | −0.02–0.16 | | 0.0972 | | 0.143 |
| | 0.17–0.27 | 0.1597 | 0.1590 | 0.1877 | 0.1957 |
| | 0.28–0.33 | | 0.2591 | | 0.2501 |
| | 0.34–0.50 | | 0.4258 | | 0.3182 |
| MNDWI | −0.60–(−0.11) | | 0.6080 | | 0.5584 |
| | −0.12–0 | 0.1105 | 0.2721 | 0.1338 | 0.3446 |
| | 0–0.23 | | 0.1199 | | 0.097 |
| TRI | 0–0.78 | | 0.4162 | | 0.2949 |
| | 0.79–1.96 | | 0.2618 | | 0.2473 |
| | 1.97–4.32 | 0.0755 | 0.1611 | 0.1025 | 0.1979 |
| | 4.33–9.82 | | 0.0986 | | 0.1502 |
| | 9.83–50.12 | | 0.0624 | | 0.1098 |
| SL | 0–2.0 | | 0.5813 | | 0.4803 |
| | >2.1–6.0 | 0.0512 | 0.3092 | 0.0594 | 0.3052 |
| | >6.0 | | 0.1096 | | 0.2145 |
| NDBI | −0.39–(−0.16) | | 0.3913 | | 0.2487 |
| | −0.15–(−0.09) | | 0.2572 | | 0.2225 |
| | −0.08–(−0.03) | 0.0358 | 0.1691 | 0.0229 | 0.1976 |
| | −0.02–(−0.01) | | 0.1100 | | 0.1693 |
| | 0.00–0.32 | | 0.0724 | | 0.1620 |

Figure 5 compares consistency ratios (CRs) between the conventional AHP and FE-AHP for the pairwise comparison matrix of thematic parameters and the sub-criteria. It is recognized that the CR should be greater than 0 and less than 0.1. A lower CR is generally preferred. A lower CR signifies a higher level of consistency in the judgments, indicating that the decision-maker's preferences are more dependable and logical. Conversely, a higher CR suggests more significant inconsistency, implying possible conflicts or contradictions in the decision-maker's preferences. Figure 5 shows that results from the FE-AHP method exhibit generally lower CR values than those obtained from the AHP method, indicating that the FE-AHP method holds a higher level of consistency and reliability. The following sections focus on the results and discussion of the maps for the thematic parameters in the study.

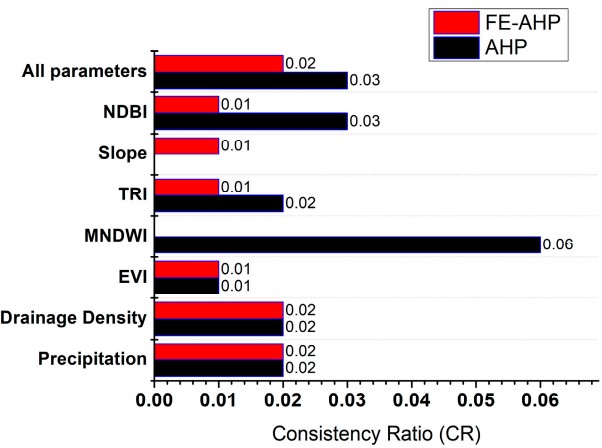

**Figure 5.** The consistency ratios (CRs) of AHP and FE-AHP for the pairwise comparison matrix.

### 3.2.1. Precipitation (P)

Figure 6 shows the year-averaged distribution of the precipitation in the CRGB. In general, the distribution of the precipitation is consistent with the land surface elevation of the groundwater basin, varying from the coastal area in the east to the mountain area in the east (see Figure 1). The considerable precipitation variation in space has made the proximal-fan an excellent recharge zone for the aquifer system. According to previous investigations, the areas with lower (or higher) precipitation were assigned a lower (or higher) rating value [50,51]. Tables A1 and A2 show the detailed calculation of the weightings and ratings [11].

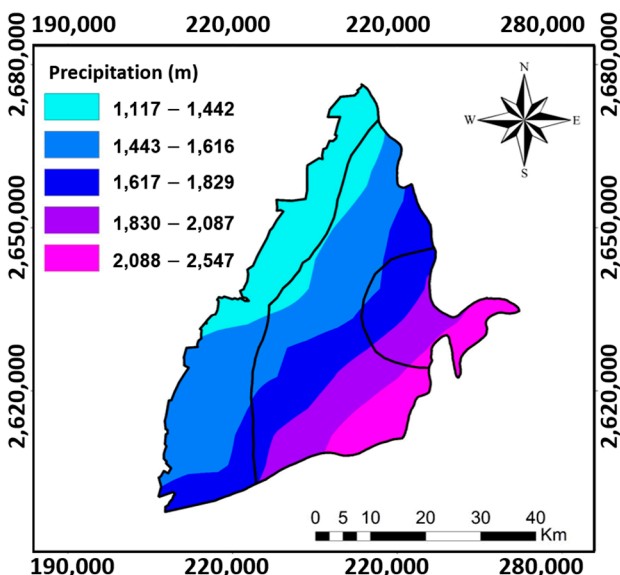

**Figure 6.** The year-averaged precipitation observed based on the CHIRPS data (PCS: UTM zone 51N).

### 3.2.2. Drainage Density (DD)

Figure 7 illustrates the drainage density results for a 1 km by 1 km cell size in the crucial agricultural regions of Western Taiwan. The well-developed drainage system, designed for agricultural purposes, includes lined channels to prevent surface water leakage. Higher drainage density correlates with increased surface runoff, potentially impacting groundwater potential negatively. Detailed calculations for the drainage density weighting and ratings in the AHP and FE-AHP methods are listed in Tables A3 and A4, respectively.

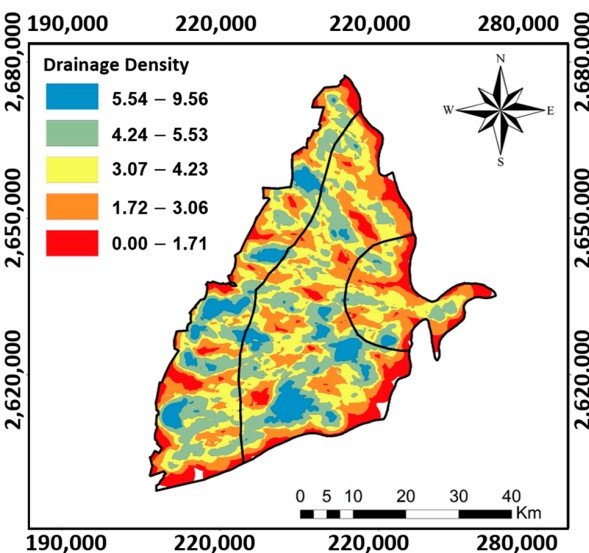

**Figure 7.** The drainage density (km/km²) derived by the DEM SRTM (PCS: UTM zone 51N).

### 3.2.3. Enhanced Vegetation Index (EVI)

Figure 8 displays the distribution of the EVI in the study area, revealing that the southeastern part of the CRGB exhibits denser vegetation cover. This denser vegetation suggests a heightened likelihood of groundwater potential in that specific region (Figure 8). For a comprehensive understanding of the assessment methodologies, detailed weightings and ratings for the AHP and FE-AHP methods can be found in Tables A5 and A6 in Appendix A.

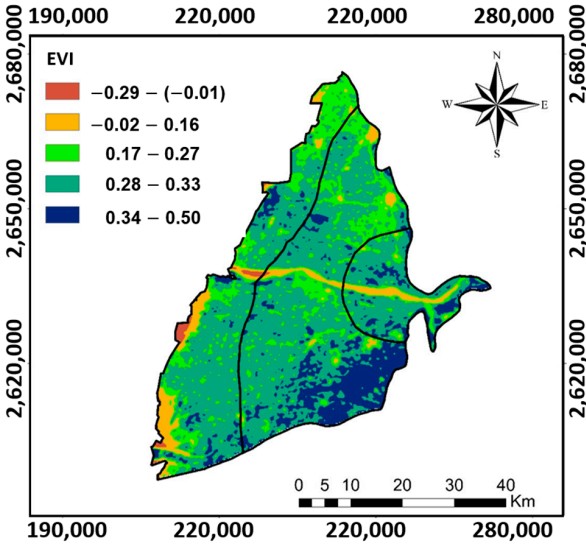

**Figure 8.** The enhanced vegetation index (EVI) calculated for the study area (PCS: UTM zone 51N).

### 3.2.4. The Modified Normalized Difference Water Index (MNDWI)

Figure 9 depicts the MNDWI results for the study area, where a high and positive MNDWI value indicates a likelihood of groundwater potential, representing the wetness of an area. However, the MNDWI cannot differentiate between salty and freshwater bodies, as it relies on spectral characteristics. Notably, the Mailiao Haipu New Land in the western part of Yunlin County features diverse land uses, including coastal uplift rejuvenation, siltation sites, saline ponds, aqua fisheries, and sandbanks affected by monsoonal activity [52]. The MNDWI can enhance groundwater potential assessment by identifying surface water bodies or wet areas, including saline ponds along the coastal line in the CRGB's

western region. Detailed weightings and ratings for the AHP and FE-AHP methods are in Tables A7 and A8 in Appendix A.

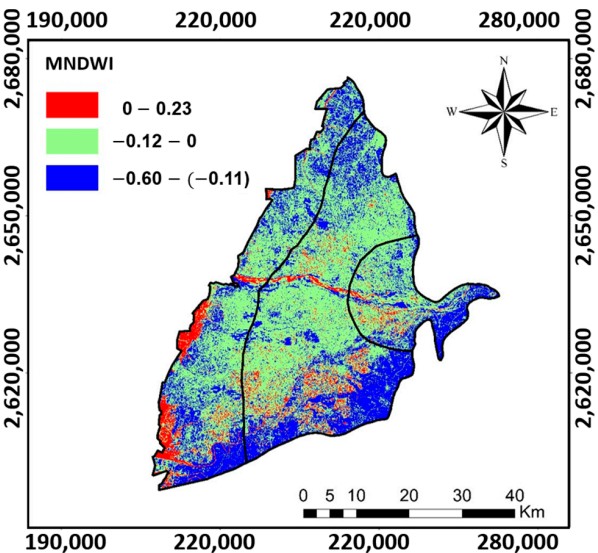

**Figure 9.** The modified normalized difference water index (MNDWI) obtained based on Landsat (PCS: UTM zone 51N).

3.2.5. Terrain Ruggedness Index (TRI)

Understanding the groundwater potential of a particular location is intricately linked to the dynamics of water bodies on the land surface and their infiltration into the aquifer system. The fate of precipitation or surface runoff is closely tied to the terrain characteristics, especially the altitude differences across the landscape. A lower terrain ruggedness index (TRI) value signifies a landscape with relatively low altitude variations, indicating a higher potential for groundwater retention [53]. This insight underscores the significance of the TRI in assessing the groundwater potential zone within the CRGB region (Figure 10). The detailed weightings and ratings of TRI for both the analytical hierarchy process (AHP) and fuzzy evaluation analytical hierarchy process (FE-AHP) methodologies can be found in Tables A9 and A10 in Appendix A, providing a comprehensive foundation for further analysis and interpretation

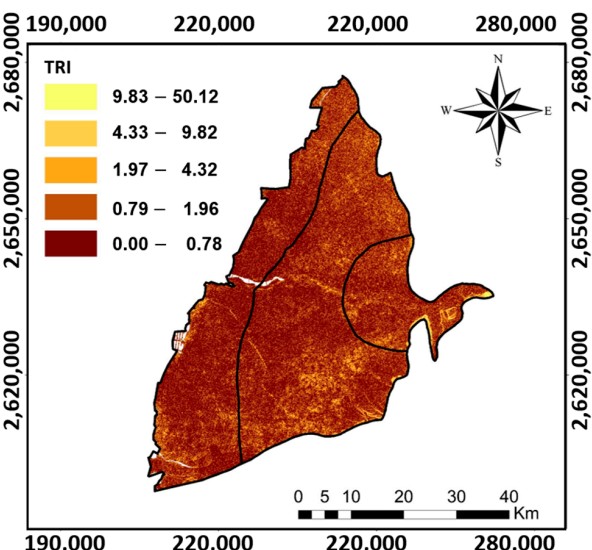

**Figure 10.** The terrain ruggedness index derived by DEM obtained from SRTM (PCS: UTM zone 51N).

3.2.6. Slope

Figure 11 shows the cell-based slope values estimated based on the DEM from the SRTM. The result reveals that the slope could vary from 0 to 30 in the study area. The CRBG is bounded by Pakuashan tableland and Douliu Hill in the east (Figure 1a). The high-slope zones are mostly in the upstream areas along the Pakuashan tableland and Douliu Hill of the CRGB. There are few spots with high slope values in the southeastern areas near the Douliu Hill. In general, low-slope areas dominate the entire CRGB, showing high infiltration potential in the CRGB. In the study, a cell with a low slope value was assigned a high weight for the AHP and FE-AHP methods. Tables A11 and A12 in Appendix A show the details of the estimated weightings for the study area. The slopes were categorized into three different intervals based on the slope values from 0.0° to 6.0°.

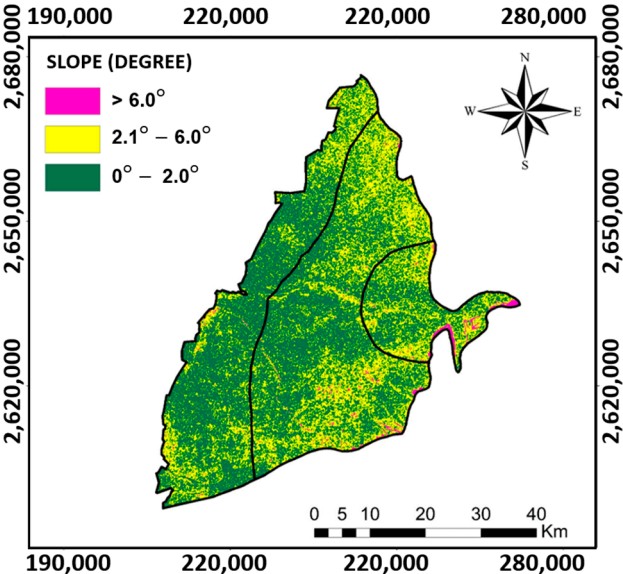

**Figure 11.** The cell-based slope values estimated by DEM SRTM (PCS: UTM zone 51N).

3.2.7. Normalized Difference Building Index (NDBI)

Figure 12 provides a comprehensive visualization of the normalized difference building index (NDBI) distribution across the CRGB region. The analysis of the NDBI reveals prominent land use types within CRGB, predominantly comprising vegetation and water bodies. Interestingly, the overall extent of built-up areas appears relatively limited, emphasizing the predominantly natural landscape of the region. This observation is crucial for understanding the interplay between land cover characteristics, particularly the scarcity of built-up areas, and their potential implications for the groundwater potential zone within the CRGB.

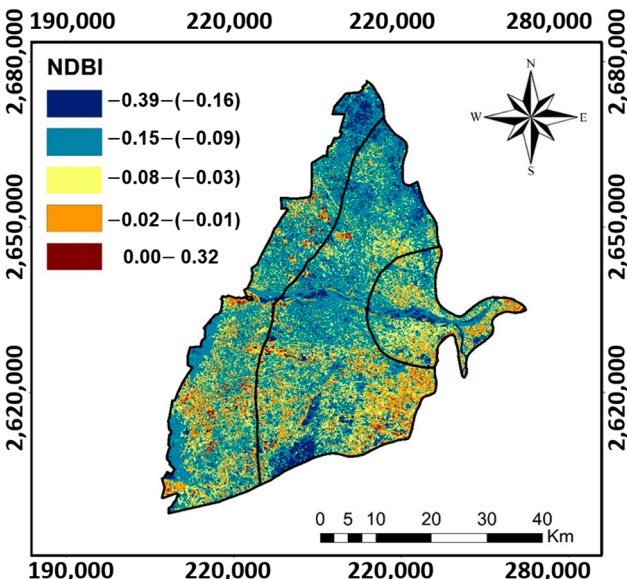

**Figure 12.** The distribution of NDBI estimated based on Landsat data (PCS: UTM zone 51N).

*3.3. Delineation of the Groundwater Potential Zone (GWPZ)*

The GWPI values obtained from Equation (15) were the basis for obtaining the GWPZ in the CRGB. The seven thematic layers were considered to assess the groundwater potential (i.e., groundwater availability, GA) in the study area. Figure 13 shows the result using the weightings and ratings calculated from the conventional AHP and FE-AHP. The characteristics of the AHP and FE-AHP made the weightings and ratings different in terms of both the values and variations within intervals (see Table 5). Tables A15 and A16 in Appendix A show the detailed weighting matrices of the selected parameters for the AHP and FE-AHP. Based on the calculation, the groundwater potential index for the study area ranges from 0.06 to 0.39, with a standard deviation of 0.05 for AHP and FE-AHP. A natural break grouping scheme based on Jenk's optimization method was used to define the range of each class for the maps of the GWPZ.

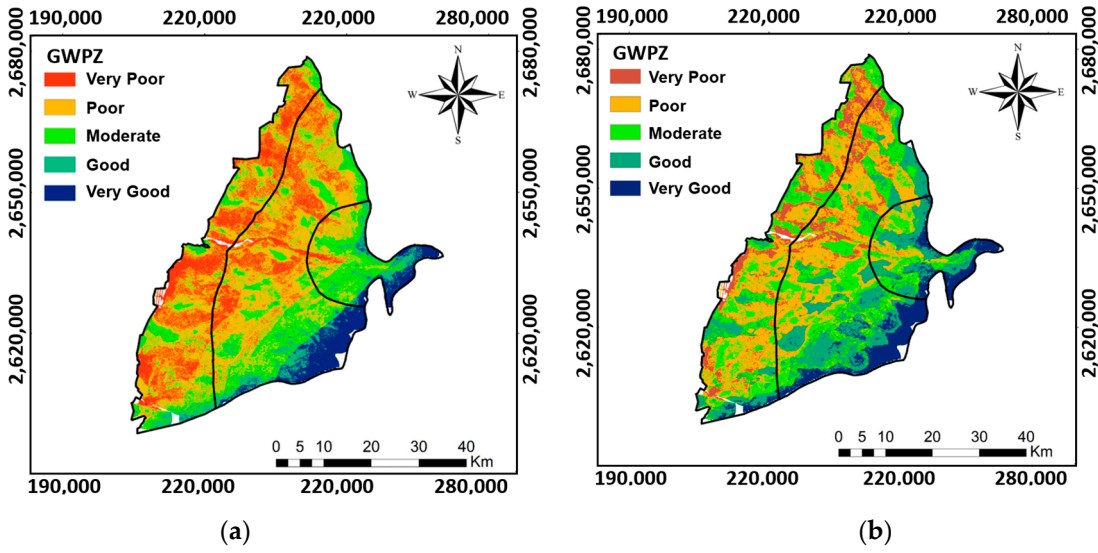

**Figure 13.** The maps of groundwater potential zone (GWPZ) for CRGB based on the results obtained from (**a**) conventional AHP and (**b**) FE-AHP methods.

The classification of the GWPZ was carried out using the AHP and FE-AHP methods, identifying five distinct groundwater potential zones, including very good, good, moderate,

poor, and very poor (Figure 13). The result obtained from the FE-AHP demonstrated a relatively smooth and reasonable variation in the GWPZ. However, the result from the AHP showed clear patterns that the precipitation distribution might control. In addition, the FE-AHP result could capture the contribution from other parameters, such as those from DD (Figure 7), EVI (Figure 8), and MNDWI (Figure 9). These parameters are essential features for the mid-fan and distal-fan areas in the CRGB. The general behavior shows that the most favorable groundwater potential zones, classified as "good" and "very good", are predominantly concentrated in the mid-fan and proximal-fan, respectively. The mid-fan and proximal-fan are indicated as areas closest to excellent potential for groundwater resources (Figure 13). The regions at the proximal-fan of the Choushui River's alluvial fan exhibit notably high groundwater recharge potential (GRP) levels. Conversely, distal-fan areas register lower GRP levels, classified as poor and very poor. In addition, regions with a moderate GRP are primarily located along the Choushui River. Based on the study of Tsai et al. (2015), a proximal-fan mainly consists of grave deposits, which supports a higher GRP. In conclusion, there was agreement between, and relevance in, the GWPZ and GRP maps proposed by Tsai et al. (2015) [47,48].

Figure 13a,b show that the areas with groundwater potential vary across the study region. The proximal fan, an upstream area, has "very good" potential for groundwater in the CRGB region. The "very good" potential is linked to abundant rainfall, sparse drainage paths, higher dense vegetation covers, rougher terrain, and steeper slopes, excluding the presence of water bodies. The lack of water bodies in a high-elevation area indicates a scarcity of surface water. Such behavior can also reduce surface water loss, increase infiltration, and create favorable hydrogeological conditions for groundwater recharge [38]. These factors collectively support higher groundwater potential in such areas. More rainfall indicates that more water recharges the groundwater. With fewer natural drainages, water can seep and recharge the groundwater reservoir more effectively. Plants play a vital role in boosting groundwater potential by using water and letting it infiltrate the ground. Rougher landscapes could slow surface water movement, allowing more time for water to penetrate and recharge the groundwater. These combined features signify areas with promising groundwater potential.

In the mid-fan, the "good" category of groundwater potential results from moderate-to-low precipitation, ensuring gradual surface water retention. The drainage density is also moderate-to-low, indicating fewer pathways for water to drain quickly. In addition, moderate-to-very-dense vegetation cover reduces surface runoff through evapotranspiration, aiding water infiltration into the soil. The area's moderate-to-high wetness indicates ample soil moisture, supporting gradual groundwater recharge. The smoother terrain and flatter slopes further slow surface water flow, giving water more time to penetrate and recharge the groundwater table. These combined factors create a balanced environment for effective groundwater replenishment without excessive water loss. These favorable conditions enhance the potential for sustained groundwater replenishment, contributing to the overall "good" groundwater potential in the area. The distal-fan area stands out as having predominantly very poor groundwater potential. The result is characterized by various factors that contribute to this classification. Firstly, this region receives the lowest precipitation levels, limiting the amount of water that can infiltrate the ground and recharge the groundwater. In addition, the area exhibits the highest drainage density, implying an extensive network of natural pathways for water to drain away and reduce groundwater recharge. In the study, the total weightings for the precipitation and drainage density dominate the calculation of the GWPZ. The vegetation and water body indexes show a relatively limited contribution to the overall GWPZ.

Figure 14 shows the distribution of the areas for different classifications of GWPZs in the CRGB. The comparison highlights the AHP's tendency to provide more distinct categorizations than the FE-AHP. The categorization of GWPZ areas indicates that the FE-AHP achieves a more realistic and balanced distribution across groundwater potential categories due to its integration of fuzzy logic and linguistic variables. When observing

the distribution range, the total area for each GWPZ category based on the FE-AHP appears to exhibit a more realistic and balanced representation across groundwater potential categories than the AHP (see Figure 14). This distribution range implies that the FE-AHP provides a broader representation of categories by effectively integrating fuzzy logic and linguistic variables.

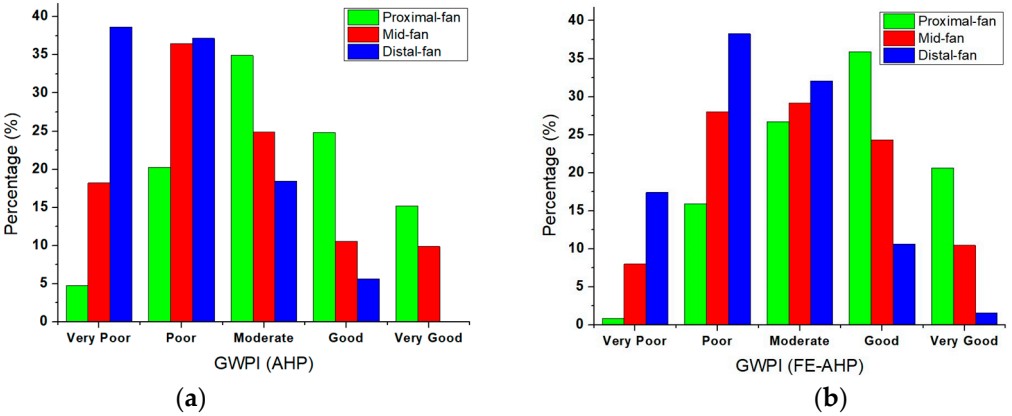

**Figure 14.** Percentage of area for the basin-fan in CRGB based on (**a**) conventional AHP and (**b**) FE-AHP methods. The green, red and blue graph are for the proximal-fan, mid-fan and distal-fan, respectively.

In summary, the result of the FE-AHP method lies in its specific feature to handle uncertainty, incorporate linguistic variables, offer a broader representation of categories, and allow for flexible weightings. These combined aspects position the FE-AHP as a powerful tool for groundwater potential assessment. This advanced approach enhances the FE-AHP to produce a smoother, more realistic, and well-balanced distribution of groundwater potential compared to the traditional AHP method. A smoother, more realistic, and well-balanced distribution of GWPZ could improve benefit judgment for groundwater management based on remote sensing data.

*3.4. Performance of the GWPI*

Table 6 shows the Pearson correlation matrix between the GWPI (groundwater potential index) and direct measurements of the GA (groundwater availability). In Table 6, the Pearson's correlation coefficient between the GWPI and GA for the AHP and FE-AHP was determined to be 0.56 (moderate correlation) and 0.67 (high correlation), respectively, which indicates a strong positive correlation for the GWPI of the FE-AHP and conventional AHP (i.e., the correlation r > 0.50) [54]. The overall correlation between the GWPI and GA showed that the FE-AHP method obtains a better match of GA than the conventional AHP method. Furthermore, the associated P-value was found to be less than 0.01, providing further evidence of the statistical significance of the correlation. This strong correlation underscores the relationship between the GWPI and groundwater availability. The high correlation between the GWPI and GA also proves the concept that remote sensing data applied to modified index–overlay approaches could provide efficient estimations of groundwater potential for shallow aquifers.

**Table 6.** Pearson's correlation matrix of GA, GWPI for AHP, and GWPI for FE-AHP.

| Layers | GA | GWPI (AHP) | GWPI (FE-AHP) |
|:---:|:---:|:---:|:---:|
| GA | 1.00 | 0.56 | 0.67 |
| GWPI (AHP) | 0.56 | 1.00 | 0.56 |
| GWPI (FE-AHP) | 0.67 | 0.88 | 1.00 |

An area with a higher GWPI value often exhibits a greater capacity to recharge and store groundwater, thus correlating with higher groundwater availability. The GWPI acts

as an indicator of groundwater potential, suggesting areas with potential for sustainable groundwater replenishment. However, it is important to acknowledge local variations in this relationship based on the hydrogeological conditions of a specific area. For instance, an area might display a high GWPI due to favorable surface characteristics, but if the underlying aquifer is shallow or overexploited, the actual groundwater availability might be limited. Understanding this relationship is crucial for effective groundwater resource management, aiding in identifying areas with significant groundwater potential and sustainably aligning groundwater usage to ensure long-term availability and prevent overexploitation.

Figure 15 shows the selected sites to check the results obtained from the FE-AHP method. The aquaculture farms along the west coast of the CRGB area are indicated as lying in the "very poor" class of GWPZ because of the low precipitation, relatively high drainage density, and low vegetation. Our results show large water bodies distributed along the coastal line on the western side of the CRGB. In the CRGB, the "poor" class of GWPZ is primarily found in areas covered by bare soil, built-up structures, and artificial materials. These land use features could decrease groundwater infiltration into the aquifer. On the other hand, the "moderate", "good", and "very good" classes of GWPZ are closely associated with the presence of ponds, minimal built-up areas, and higher vegetation cover, which contribute to groundwater recharge. Ground checking confirms consistency between the GWPZ map and land use.

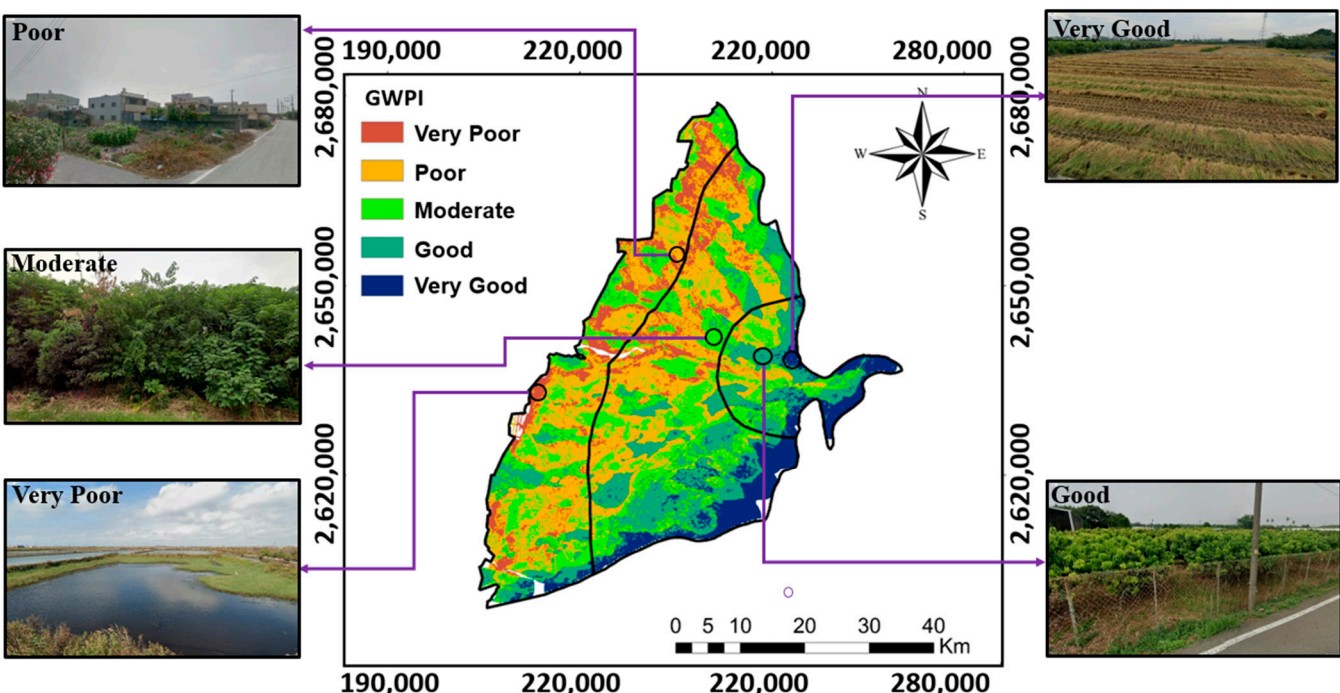

**Figure 15.** Field surveys for different classes of GWPZs at selected sites using images from Google Street View. The purple line corresponds to distinct locations defined by each class of GWPZ, while the black line represents the proximal-fan, mid-fan, and distal-fan categories as illustrated in Figure 1.

## 4. Conclusions

This study integrated the normalized index–overlay method and the fuzzy extended analytical hierarchy process (FE-AHP) to map cost-effective GWPZs. The proposed normalized index–overlay method collects the weightings and ratings of several prospective thematic layers. A more realistic approach to weighting parameters is achieved by normalizing and correlating the parameters with observed groundwater availability (GA) as a baseline for ranking parameter layers. The observed GA was calculated based on site-specific observations such as the aquifer thickness, depth to the groundwater level, and porosity of the aquifer materials. Seven comprehensive thematic layers from remote

sensing (RS) data were processed to obtain the weightings and ratings of the groundwater potential index (GWPI) for the Choushui River groundwater basin (CRGB) in Western Taiwan. In the study, the selection of parameters was based on hydrological processes, human interventions, the geological profile, and the surface profile. Hydrological processes were characterized by specific parameters such as P (precipitation in mm), MNDWI (modified normalized difference water index), and DD (drainage density in $km/km^2$). Human interventions were identified by two parameters, namely, the EVI (enhanced vegetation index) and NDBI (normalized difference building index). The geological profile and surface profile were explained by the TRI (terrain ruggedness index) and SL (slope in degrees), respectively.

In the study, the conventional AHP method was employed for comparison purposes in the mapping of the GWPZ. FE-AHP could yield smooth thresholds of each selected class in specific factors. The site chosen to prove the proposed concept in the study was the CRGB. The AHP and FE-AHP results show that the estimated groundwater potential map matched the groundwater availability derived from the direct measurements field data. The FE-AHP method demonstrated more reasonable weight distribution and smoother transitions in ratings, offering a refined and consistent decision-making process. Based on consistency ratios (CRs), the FE-AHP method consistently showed lower CR values than the AHP method, indicating higher reliability and consistency in decision-making preferences. The proposed approach could be useful for efficiently estimating groundwater availability in a shallow aquifer system, in which groundwater monitoring data and aquifer properties are insufficient or unavailable.

The groundwater potential varies in the CRGB region. The proximal-fan, upstream, boasts "very good" potential due to abundant rainfall, low drainage density, dense vegetation, rough terrain, and steep slopes, excluding water bodies. The absence of water bodies at higher elevations implies less surface water loss and more groundwater recharge. Lower drainage density facilitates effective seepage and recharge. In the mid-fan, "good" groundwater potential results from moderate-to-low precipitation, allowing gradual absorption. The distal-fan area predominantly shows "very poor" groundwater potential due to low precipitation, high drainage density impeding recharge, limited vegetation, and a minimal contribution of water bodies due to possible salinity.

The GWPZ map delineates five distinct groundwater potential zones. According to the GWPI from the AHP and FE-AHP ("moderate" to "very good" categories), about 49.00% and 59.56% of the CRGB area can be categorized as potential groundwater recharge zones, respectively. In summary, the advantage of the FE-AHP method lies in its capacity to handle uncertainty, integrate linguistic variables, provide a broader representation of categories, and allow for flexible weighting. Therefore, the FE-AHP method could produce a smoother, more realistic, and more well-balanced distribution of groundwater potential compared to the conventional AHP method.

The findings of this study establish an initial framework for understanding the groundwater potential zones in the CRGB, laying the foundation for sustainable groundwater resource management in the basin. Given the generalizable characteristics and logical conditions employed in this approach, it can be adapted and implemented in other regions with necessary adjustments. The results of the GWPZ map can serve as valuable guidelines for future planning endeavors related to sustainable groundwater recharge in the CRGB. These insights empower policymakers to make informed decisions regarding groundwater resource management.

In this study, the remote sensing data did not directly obtain the key aquifer properties such as the thickness, hydraulic conductivity, and types of soil material. All these site-specific observations could be costly because of the time and resources devoted to them. Additional remote sensing technologies could be useful to improve the analysis of groundwater potential. Further validation of this approach could be conducted through monitoring groundwater well discharge and step drawdown pumping well tests at various

CRGB locations. Such tests would evaluate specific yields across different GWPZs, allowing a thorough examination of groundwater resources in shallow aquifer systems.

**Author Contributions:** L.N. and C.-F.N. conceived and designed the research, providing valuable insights during its conception. I.-H.L., C.-P.L. and N.H.H. collected and analyzed the hydrogeological data for the study area. Y.D. contributed to the preparation and processing of remote sensing data. L.N. conducted the study, carried out the analyses, and drafted the initial manuscript. C.-F.N. and W.-C.L. further refined and finalized the manuscript for communication with the journal. All authors have read and agreed to the published version of the manuscript.

**Funding:** This research was partially supported by the National Science and Technology Council, the Republic of China, under grants NSTC 111-2621-M-008-003, NSTC 112-MOEA-M-008-001, NSTC 112-2123-M-008-001, and NSTC 112-2122-M-007-002.

**Data Availability Statement:** The data presented in this study are available in article.

**Acknowledgments:** The authors thank the Taiwan Water Resource Agency (WRA) and Central Geological Survey (CGS) for the groundwater data, precipitation data, hydrogeological loggings, and lithology data.

**Conflicts of Interest:** The authors declare no conflicts of interest.

## Appendix A

**Table A1.** Comparison matrix and significance weighting values of precipitation (AHP).

|  | 1117–1442 | 1443–1616 | 1617–1829 | 1830–2087 | 2088–2547 | Weight |
|---|---|---|---|---|---|---|
| 1117–1442 | 1.00 | 0.50 | 0.25 | 0.20 | 0.14 | 0.0488 |
| 1443–1616 | 2.00 | 1.00 | 0.50 | 0.25 | 0.20 | 0.0817 |
| 1617–1829 | 4.00 | 2.00 | 1.00 | 0.50 | 0.25 | 0.1491 |
| 1830–2087 | 5.00 | 4.00 | 2.00 | 1.00 | 0.50 | 0.2667 |
| 2088–2547 | 7.00 | 5.00 | 4.00 | 2.00 | 1.00 | 0.4537 |

**Table A2.** Comparison matrix and significance weighting values of precipitation (FE-AHP).

|  | 1117–1442 | 1443–1616 | 1617–1829 | 1830–2087 | 2088–2547 | Weight |
|---|---|---|---|---|---|---|
| 1117–1442 | (1, 1, 1) | (2/3, 1, 2) | (2/5, 1/2, 2/3) | (1/3, 2/5, 1/2) | (2/7, 1/3, 2/5) | 0.0171 |
| 1443–1616 | (1/2, 1, 3/2) | (1, 1, 1) | (2/3, 1, 2) | (2/5, 1/2, 2/3) | (1/3, 2/5, 1/2) | 0.0909 |
| 1617–1829 | (3/2, 2, 5/2) | (1/2, 1, 3/2) | (1, 1, 1) | (2/3, 1, 2) | (2/5, 1/2, 2/3) | 0.2008 |
| 1830–2087 | (2, 5/2, 3) | (3/2, 2, 5/2) | (1/2, 1, 3/2) | (1, 1, 1) | (2/3, 1, 2) | 0.3042 |
| 2088–2547 | (5/2, 3, 7/2) | (2, 5/2, 3) | (3/2, 2, 5/2) | (1/2, 1, 3/2) | (1, 1, 1) | 0.3870 |

**Table A3.** Pairwise comparison matrix and significance weighting of drainage density (AHP).

|  | 0–1.71 | 1.72–3.06 | 3.07–4.23 | 4.24–5.54 | 5.55–9.56 | Weight |
|---|---|---|---|---|---|---|
| 0–1.71 | 1.00 | 2.00 | 4.00 | 5.00 | 6.00 | 0.4456 |
| 1.72–3.06 | 0.50 | 1.00 | 2.00 | 4.00 | 5.00 | 0.2690 |
| 3.07–4.23 | 0.25 | 0.50 | 1.00 | 2.00 | 4.00 | 0.1512 |
| 4.24–5.54 | 0.20 | 0.25 | 0.50 | 1.00 | 2.00 | 0.0827 |
| 5.55–9.56 | 0.17 | 0.20 | 0.25 | 0.50 | 1.00 | 0.0514 |

**Table A4.** Pairwise comparison matrix and significance weighting of drainage density (FE-AHP).

|  | 0–1.71 | 1.72–3.06 | 3.07–4.23 | 4.24–5.54 | 5.55–9.56 | Weight |
|---|---|---|---|---|---|---|
| 0–1.71 | (1,1, 1) | (1/2, 1, 3/2) | (3/2, 2, 5/2) | (2, 5/2, 3) | (5/2, 3, 7/2) | 0.3870 |
| 1.72–3.06 | (2/3, 1, 2) | (1,1, 1) | (1/2, 1, 3/2) | (3/2, 2, 5/2) | (2, 5/2, 3) | 0.3042 |
| 3.07–4.23 | (2/5, 1/2, 2/3) | (2/3, 1, 2) | (1,1, 1) | (1/2, 1, 3/2) | (3/2, 2, 5/2) | 0.2008 |
| 4.24–5.54 | (1/3, 2/5, 1/2) | (2/5, 1/2, 2/3) | (2/3, 1, 2) | (1,1, 1) | (1/2, 1, 3/2) | 0.0909 |
| 5.55–9.56 | (2/7, 1/3, 2/5) | (1/3, 2/5, 1/2) | (2/5, 1/2, 2/3) | (2/3, 1, 2) | (1,1, 1) | 0.0171 |

**Table A5.** Comparison matrix and significance weighting values of EVI (AHP).

|  | −0.29–(−0.01) | −0.02–0.16 | 0.17–0.27 | 0.28–0.33 | 0.34–0.50 | Weight |
|---|---|---|---|---|---|---|
| −0.29–(−0.01) | 1.00 | 0.50 | 0.30 | 0.30 | 0.20 | 0.0588 |
| −0.02–0.16 | 2.00 | 1.00 | 0.50 | 0.30 | 0.30 | 0.0972 |
| 0.17–0.27 | 3.00 | 2.00 | 1.00 | 0.50 | 0.30 | 0.1590 |
| 0.28–0.33 | 4.00 | 3.00 | 2.00 | 1.00 | 0.50 | 0.2591 |
| 0.34–0.50 | 6.00 | 4.00 | 3.00 | 2.00 | 1.00 | 0.4258 |

**Table A6.** Comparison matrix and significance weighting values of EVI (FE-AHP).

|  | −0.29–(−0.01) | −0.02–0.16 | 0.17–0.27 | 0.28–0.33 | 0.34–0.50 | Weight |
|---|---|---|---|---|---|---|
| −0.29–(−0.01) | (1,1,1) | (2/3, 1, 2) | (1/2, 2/3, 1) | (2/5, 1/2, 2/3) | (2/7, 1/3, 2/5) | 0.0930 |
| −0.02–0.16 | (1/2, 1, 3/2) | (1,1,1) | (2/3, 1, 2) | (1/2, 2/3, 1) | (2/5, 1/2, 2/3) | 0.1430 |
| 0.17–0.27 | (1, 3/2, 2) | (1/2, 1, 3/2) | (1,1,1) | (2/3, 1, 2) | (1/2, 2/3, 1) | 0.1957 |
| 0.28–0.33 | (3/2, 2, 5/2) | (1, 3/2, 2) | (1/2, 1, 3/2) | (1,1,1) | (2/3, 1, 2) | 0.2501 |
| 0.34–0.50 | (5/2, 3, 7/2) | (3/2, 2, 5/2) | (1, 3/2, 2) | (1/2, 1, 3/2) | (1,1,1) | 0.3182 |

**Table A7.** Comparison matrix and significance weighting values of MNDWI (AHP).

|  | −0.60–(−0.11) | −0.12–0 | 0–0.23 | Weight |
|---|---|---|---|---|
| −0.60–(−0.11) | 1.00 | 3.00 | 4.00 | 0.6080 |
| −0.12–0 | 0.30 | 1.00 | 3.00 | 0.2721 |
| 0–0.23 | 0.30 | 0.30 | 1.00 | 0.1199 |

**Table A8.** Comparison matrix and significance weighting values of MNDWI (FE-AHP).

|  | −0.60–(−0.11) | −0.12–0 | 0–0.23 | Weight |
|---|---|---|---|---|
| −0.60–(−0.11) | (1, 1, 1) | (1, 3/2, 2) | (3/2, 2, 5/2) | 0.5584 |
| −0.12–0 | (1/2, 2/3, 1) | (1, 1, 1) | (1, 3/2, 2) | 0.3446 |
| 0–0.23 | (2/5, 1/2, 2/3) | (1/2, 2/3, 1) | (1, 1, 1) | 0.0970 |

**Table A9.** Comparison matrix and significance weighting values of TRI (AHP).

|  | 0–0.78 | 0.79–1.96 | 1.97–4.32 | 4.33–9.82 | 9.83–50.12 | Weight |
|---|---|---|---|---|---|---|
| 0–0.78 | 1.00 | 2.00 | 3.00 | 4.00 | 5.00 | 0.4162 |
| 0.79–1.96 | 0.50 | 1.00 | 2.00 | 3.00 | 4.00 | 0.2618 |
| 1.97–4.32 | 0.30 | 0.50 | 1.00 | 2.00 | 3.00 | 0.1611 |
| 4.33–9.82 | 0.30 | 0.30 | 0.50 | 1.00 | 2.00 | 0.0986 |
| 9.83–50.12 | 0.20 | 0.30 | 0.30 | 0.50 | 1.00 | 0.0624 |

**Table A10.** Comparison matrix and significance weighting values of TRI (FE-AHP).

|  | 0–0.78 | 0.79–1.96 | 1.97–4.32 | 4.33–9.82 | 9.83–50.12 | Weight |
|---|---|---|---|---|---|---|
| 0–0.78 | (1,1,1) | (1/2, 1, 3/2) | (1, 3/2, 2) | (3/2, 2, 5/2) | (2, 5/2, 3) | 0.2949 |
| 0.79–1.96 | (2/3, 1, 2) | (1,1,1) | (1/2, 1, 3/2) | (1, 3/2, 2) | (3/2, 2, 5/2) | 0.2473 |
| 1.97–4.32 | (1/2, 2/3, 1) | (2/3, 1, 2) | (1,1,1) | (1/2, 1, 3/2) | (1, 3/2, 2) | 0.1979 |
| 4.33–9.82 | (2/5, 1/2, 2/3) | (1/2, 2/3, 1) | (2/3, 1, 2) | (1,1,1) | (1/2, 1, 3/2) | 0.1502 |
| 9.83–50.12 | (1/3, 2/5, 1/2) | (2/5, 1/2, 2/3) | (1/2, 2/3, 1) | (2/3, 1, 2) | (1,1,1) | 0.1098 |

**Table A11.** Comparison matrix and significance weighting values of slope (AHP).

|  | 0–2.0° | 2.1–6.0° | >6.0° | Weight |
|---|---|---|---|---|
| 0–2.0° | 1.00 | 2.00 | 5.00 | 0.5813 |
| 2.1–6.0° | 0.50 | 1.00 | 3.00 | 0.3092 |
| >6.0° | 0.20 | 0.30 | 1.00 | 0.1096 |

**Table A12.** Comparison matrix and significance weighting values of slope (FE-AHP).

|  | **0–2.0°** | **2.1–6.0°** | **>6.0°** | **Weight** |
|---|---|---|---|---|
| 0–2.0° | (1,1, 1) | (1/2, 1, 3/2) | (2, 5/2, 3) | 0.4803 |
| 2.1–6.0° | (1/2, 2/3, 1) | (1,1, 1) | (1, 3/2, 2) | 0.3052 |
| >6.0° | (1/3, 2/5, 1/2) | (2/3, 1, 2) | (1,1, 1) | 0.2145 |

**Table A13.** Comparison matrix and significance weighting values of NDBI (AHP).

|  | **−0.39–(−0.16)** | **−0.15–(−0.09)** | **−0.08–(−0.03)** | **−0.02–(−0.01)** | **0.00–0.32** | **Weight** |
|---|---|---|---|---|---|---|
| −0.39–(−0.16) | 1.00 | 2.00 | 3.00 | 3.00 | 4.00 | 0.3913 |
| −0.15–(−0.09) | 0.50 | 1.00 | 2.00 | 3.00 | 3.00 | 0.2572 |
| −0.08–(−0.03) | 0.30 | 0.50 | 1.00 | 2.00 | 3.00 | 0.1691 |
| −0.02–(−0.01) | 0.30 | 0.30 | 0.50 | 1.00 | 2.00 | 0.1100 |
| 0.00–0.32 | 0.30 | 0.30 | 0.30 | 0.50 | 1.00 | 0.0724 |

**Table A14.** Comparison matrix and significance weighting values of NDBI (FE-AHP).

|  | **−0.39–(−0.16)** | **−0.15–(−0.09)** | **−0.08–(−0.03)** | **−0.02–(−0.01)** | **0.00–0.32** | **Weight** |
|---|---|---|---|---|---|---|
| −0.39–(−0.16) | (1,1, 1) | (1/2, 1, 3/2) | (1, 3/2, 2) | (1, 3/2, 2) | (3/2, 2, 5/2) | 0.2487 |
| −0.15–(−0.09) | (2/3, 1, 2) | (1, 1, 1) | (1/2, 1, 3/2) | (1, 3/2, 2) | (1, 3/2, 2) | 0.2225 |
| −0.08–(−0.03) | (1/3, 2/3, 1) | (2/3, 1, 2) | (1,1, 1) | (1/2, 1, 3/2 | (1, 3/2, 2) | 0.1976 |
| −0.02–(−0.01) | (1/3, 2/3, 1) | (1/2, 2/3, 1) | (2/3, 1, 2) | (1,1, 1) | (1/2, 1, 3/2) | 0.1693 |
| 0.00–0.32 | (2/5, ½, 2/3) | (1/2, 2/3, 1) | (1/2, 2/3, 1) | (2/3, 1, 3) | (1,1, 1) | 0.1620 |

**Table A15.** Comparison matrix and significance weighting values of using parameters (AHP).

|  | **Precipitation** | **Drainage Density** | **EVI** | **MNDWI** | **TRI** | **Slope** | **NDBI** | **Weight** |
|---|---|---|---|---|---|---|---|---|
| Precipitation | 1.00 | 0.50 | 0.33 | 0.25 | 0.20 | 0.17 | 0.14 | 0.3354 |
| Drainage Density | 2.00 | 1.00 | 0.50 | 0.33 | 0.25 | 0.20 | 0.17 | 0.2320 |
| EVI | 3.00 | 2.00 | 1.00 | 0.50 | 0.33 | 0.25 | 0.20 | 0.1597 |
| MNDWI | 4.00 | 3.00 | 2.00 | 1.00 | 0.50 | 0.33 | 0.25 | 0.1105 |
| TRI | 5.00 | 4.00 | 3.00 | 2.00 | 1.00 | 0.50 | 0.33 | 0.0755 |
| Slope | 6.00 | 5.00 | 4.00 | 3.00 | 2.00 | 1.00 | 0.50 | 0.0755 |
| NDBI | 7.00 | 6.00 | 5.00 | 4.00 | 3.00 | 2.00 | 1.00 | 0.0512 |

**Table A16.** Comparison matrix and significance weighting values of using parameters (FE-AHP).

|  | **Precipitation** | **Drainage Density** | **EVI** | **MNDWI** | **TRI** | **Slope** | **NDBI** | **Weight** |
|---|---|---|---|---|---|---|---|---|
| Precipitation | (1, 1, 1) | (1/2, 1, 3/2) | (1, 3/2, 2) | (3/2, 2, 5/2) | (3/2, 2, 5/2) | (2, 5/2, 3) | (5/2, 3, 7/2) | 0.2661 |
| Drainage Density | (2/3, 1, 2) | (1, 1, 1) | (1/2, 1, 3/2) | (1, 3/2, 2) | (3/2, 2, 5/2) | (3/2, 2, 5/2) | (2, 5/2, 3) | 0.2277 |
| EVI | (1/2, 2/3, 1) | (2/3, 1, 2) | (1, 1, 1) | (1/2, 1, 3/2) | (1, 3/2, 2) | (3/2, 2, 5/2) | (3/2, 2, 5/2) | 0.1877 |
| MNDWI | (2/5, 1/2, 2/3) | (1/2, 2/3, 1) | (2/3, 1, 2) | (1, 1, 1) | (1/2, 1, 3/2) | (1, 3/2, 2) | (3/2, 2, 5/2) | 0.1338 |
| TRI | (2/5, 1/2, 2/3) | (2/5, 1/2, 2/3) | (1/2, 2/3, 1) | (2/3, 1, 2) | (1, 1, 1) | (1/2, 1, 3/2) | (1, 3/2, 2) | 0.1025 |
| Slope | (1/3, 2/5, 1/2) | (2/5, 1/2, 2/3) | (2/5, 1/2, 2/3) | (1/2, 2/3, 1) | (2/3, 1, 2) | (1, 1, 1) | (1/2, 1, 3/2) | 0.0594 |
| NDBI | (2/7, 1/3, 2/5) | (1/3, 2/5, 1/2) | (2/5, 1/2, 2/3) | (2/5, 1/2, 2/3) | (1/2, 2/3, 1) | (2/3, 1, 2) | (1, 1, 1) | 0.0229 |

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
