# Peer review of "Cost-Effective Groundwater Potential Mapping by Integrating Multiple Remote Sensing Data and the Index–Overlay Method"

_remotesensing, doi:10.3390/rs16030502_

Round 1

Reviewer 1 Report

Comments and Suggestions for Authors

Authors of the paper use a variety of remote sensing-based datasets and integrates the normalized index-overlay method and the fuzzy extended-analytical hierarchy process to map cost-effective GWPZs.

The aim of the article (“…providing significant insight into the strategies of the groundwater management plan …” [lines 44-45]) is of interest to the typical reader of the MDPI Remote Sensing journal if the authors can fix the shortcomings of the original manuscript. The work is interesting and can be published after a major revision with the following comments: 

[1] The abstract should provide more specific conclusions, highlighting the innovative aspects of the work.

[2] The longitude and latitude of Figure 1(a) only need to be given in degrees, and the image resolution is too low. And the font is not clear, and there is image masking.

[3] (line 132) ‘the Analytic Hierarchy Process (AHP) method has seen extensive application’ lacks necessary literature citations.

[4] (line 166-170) There is no need for an introduction to the structure of the article.

[5] (lines 230) section should be titled ‘Data Sets 

[6] (line 231-250) Can more detailed information be provided, such as the temporal resolution of the data and the website from which the data can be obtained?

[7] (throughout the manuscript) several acronyms (e.g., AHP {line 310}) are defined in multiple locations throughout the manuscript 

[8] (throughout the Methodology section) every variable of each equation must be defined, along with listing their appropriate units (e.g., (km/km2)? {line 355})  

[9] Please place the figures in appropriate locations, around the paragraphs that reference them. (e.g., Figure 3a {line 409})  

[10] (line 519-520) Please indicate the selected locations on the map and describe the reasons for selecting that particular location.

[11] In Figure 5, please display the numbers as percentages and only retain two decimal places.

[12] (line 618) ‘previous investigations’ lacks necessary literature citations. 

[13] (throughout the Results and Discussion section) There is an extensive focus on describing the methodology, while lacking further analysis and discussion of the results (e.g., 3.2.3. Enhanced vegetation index (EVI){line 643})  

[14] What is the source of information being presented in Fig. 14? Please add a legend description to the figure caption.

[15] (line 861) ‘…In Table 6, the Pearson's correlation coefficient between GWPI and GWA for AHP and FE-AHP was determined to be 0.56 and 0.67, respectively…’,correlation coefficient between GWPI and GWA? Please describe accurately. Additionally, I do not believe that an accuracy improvement from 0.56 to 0.67 is significant. Regarding the statement 'high correlation between GWPI and GA,' I do not think they have a high correlation. Can you explain the reasons for the low correlation between GWPI and the measured data GA? Is there potential for improvement in future research?"

[16] Please redraw the selected sites in Figure 15, as it is not clear. Why were fewer sites used? I don't believe that such a limited number of sites is sufficient to support the conclusions of the paper.

[17] Conclusions should be improved

Comments on the Quality of English Language

The language can be improved.

Author Response

Dear Reviewer#1,

We appreciate the time and effort that you provide valuable feedback on our manuscript. We have made changes based on the suggestions provided. We also highlighted the changes in the uploaded manuscript. The attached file includes the list of the point-by-point responses to the comments and concerns.

Thank you for the valuable comments.

Chuen-Fa Ni, Ph.D., P.E., Distinguished Professor/Director, Graduate Institute of Applied Geology, National Central University Director, Center for Environmental Studies, National Central University
No. 300, Zhongda Rd., Zhongli City, Taoyuan 32001, Taiwan.
E-mail: [email protected][email protected] Tel: +886-3-4227151 ext. 65874 Fax: +886-3-4263127

Reviewer 2 Report

Comments and Suggestions for Authors

1.The abstract is clear and informative, but it can be more concise. Try to summarize the key points more succinctly.

2.Specify the study area in the abstract. Instead of just mentioning "western Taiwan," you can mention the specific location or region to provide more context.

3. In the abstract, you can briefly mention the results or implications of the study. What are the key findings, and why are they significant?

 4. Check for grammatical issues and sentence structure. For example, "Results of the observed GWPZs are classified into five potential groundwater zones, including very good, good, moderate, poor, and very poor." This sentence is somewhat complex; consider breaking it down into more digestible segments

5. Start by providing a brief overview of the importance of groundwater availability and its relevance to the study area. Explain why it's crucial to evaluate groundwater potential in this particular region.

6. Provide more context about why the integrated approach of FE-AHP and the normalized index-overlay method was chosen. What motivated this choice, and how does it address the limitations of other methods?

Author Response

Dear Reviewer,

We appreciate the time and effort that you provide valuable feedback on our manuscript. We have made changes to reflect the suggestions provided. We also highlighted the changes in the manuscript. The attached file includes the list of the point-by-point responses to the comments and concerns.

Thank you for the valuable comments.

Chuen-Fa Ni, Ph.D., P.E., Distinguished Professor/Director, Graduate Institute of Applied Geology, National Central University Director, Center for Environmental Studies, National Central University
No. 300, Zhongda Rd., Zhongli City, Taoyuan 32001, Taiwan.
E-mail: [email protected][email protected] Tel: +886-3-4227151 ext. 65874 Fax: +886-3-4263127

Reviewer 3 Report

Comments and Suggestions for Authors

Interesting paper. Congratulations. Very long (could probably be shortened for an international readership). E.g. line 80-100 read like advertising text for remote sensing. Excessive use of abbreviations.

Check line 78: "...groundwater potential evaluations have been used for in-situ measurements" > I think it should be vice versa.

Table3 : Please check the porosity of 0.776 for Clay loam.

Figure 2: The flow diagram contains an internal loop. Is this intended? not very clear.

Overall: It is a pity that the paper just compares AHP to FE-AHP remote sensing. The research would need to be validated with the results of a conventional groundwater availibility mapping. The interconnectedness of the groundwater in the aquifer seems to be disregarded in this kind of excercise. I fear that the GWPZ-information will be given too much unreflected trust by decision makers. As such, the method seems o be over-promising. I suggest to reflect that critically in the discussion.

Author Response

(The authors gave the same response as above.)

Reviewer 4 Report

Comments and Suggestions for Authors

The groundwater availability (GA) is not only dependent on the first (surface) layer of an aquifer system as water moves from the deeper layers to the surface layer via the processes of soil matric potential, soil moisture redistribution, evapotranspiration-driven soil water movement in unsaturated zone, recharge/discharge, etc. To make this research more robust, authors need to integrate inter-connectivity of groundwater characteristics by including multilayer-based groundwater hydrology and unsaturated zone hydrology. For example, although the first (surface) layer would look to have more GA based on remote sensing (RS) data, that doesn’t necessarily mean that a specific location has a truly higher GA than other locations because groundwater is not going to be pumped out of the surface layer in most of cases. The GA estimation should be based on more than the groundwater hydrology of the surface layer, per se (Lines 287-289).

As a reviewer, I understand that this research group would like to apply a newly developed method to look for GA by utilizing remote sensing and GIS data. However, the backbone theory should make sense to get used to conduct this type of research no matter how a developed methodology would be good in any field of studies including groundwater hydrology and modeling.

To my humble opinion, authors approach to apply an integrated method to identify GA in Choushui River groundwater basin doesn’t seem to be quite reasonable due to the above reasons.

As a minor comment, this manuscript integrates the index-overlay method and fuzzy extended-analytical hierarchy process (FE-AHP), but the FE-AHP looks to be used more actively compared to the index-overlay method. So, the title could be improved after this manuscript is going to be re-written.

Comments on the Quality of English Language

Overall, it is ok. However, this manuscript can use minor editing.

Author Response

(The authors gave the same response as above.)

Reviewer 5 Report

Comments and Suggestions for Authors

remotesensing-2702950 “Cost-effective Groundwater Potential Mapping by Integrating Multiple Remote Sensing Data and the Index-overlay Method” L.Nainggolan, C-F.Ni, Y.Darmawan, WC.Lo, I-H.Lee, C-P.Lin, NH.Hiep

This article describes the use of the fuzzy mapping technique FE-AHP applied to the problem of mapping potential usable groundwater resources. It is quite long and a little repetitive to read – I hope the lengthy table sin the Appendix will become additional material rather than included in the final paper. There is adequate use of remotely sensed data, and a fair comparison between the original AHP and this FE-AHP application.

The authors argue at the end of §3.3 that “... a smoother, more realistic and well-balanced distribution of groundwater potential …” is desirable. I would argue that one that is more correct is more desirable regardless of its distribution of classes; if the entire area has low GWP then it should all be designated as such. Equal-area coloured maps may be more aesthetically pleasing to a human eye but are not necessarily the “right answer”.

Table 2 has an error in the entry for “clay loam” that should be 0.476. It is also worth noting that the values in the table, and in Clapp and Hornberger, are saturated water content (qs) and not porosity (as intimated in the original paper) so that they exaggerate available storage. It is a simple matter to estimate water content at an arbitrarily high suction, say 3MPa = 30 bar = 300m, for a residual water content (qr) and find the difference. In this case the porosity (qs-qr) for sand is 34% and it decreases down to clay with 21%. Using the values from Clapp and Hornberger will diminish the influence of soil type as a factor in GA or GWPZ, since the values indicated are all similar; mean qs=0.450 with CV=7.6%, compared to mean (qs-qr)=0.282 with CV=18.3%. Furthermore, when comparing the soils ranking by (qs-qr) to qs, there is a negative relationship (R2=0.26, significant at 10% level 1-sided), which would imply the opposite influence of available soil storage by soil type on GA.

Table 3 confuses me (or maybe I just haven’t fully understood the text yet). In Saaty’s work, the AHP scale 1-9 represents equally (relatively) important variables at “1” to extremely important variables at “9”. The fuzzy table goes from “equally strong" to "equally strong” – is there any representation of an extremely important variable in FE-AHP? The tables in the Appendices indicate that the weightings with FE-AHP are more smooth and less variable than AHP, which is highlighted in Figure 13 where some of the boundaries of categories are clearly visible in the GWPZ results for AHP.

Comments on the Quality of English Language

The English language expression is satisfactory, although the authors tend to repeat a sentence twice near the start or end of a section. For example "This technique leads to these outcomes. The outcomes required need these techniques." This should be avoided.

Author Response

Dear Reviewer,

We appreciate the time and effort that you provide valuable feedback on our manuscript. We have made changes to reflect the suggestions provided. We also highlighted the changes in the manuscript. The attached file includes the list of the point-by-point responses to the comments and concerns.

Thank you for your time.

Chuen-Fa Ni, Ph.D., P.E., Distinguished Professor/Director, Graduate Institute of Applied Geology, National Central University Director, Center for Environmental Studies, National Central University
No. 300, Zhongda Rd., Zhongli City, Taoyuan 32001, Taiwan.
E-mail: [email protected][email protected] Tel: +886-3-4227151 ext. 65874 Fax: +886-3-4263127

Round 2

Reviewer 1 Report

Comments and Suggestions for Authors

I think the revised version has been improved novelly, with all points covered.

So I suggest to accept the manuscript.  

Reviewer 2 Report

Comments and Suggestions for Authors

ACCEPT